# Differentiable Reasoning about Knowledge Graphs with Reshuffled Embeddings

## Abstract

Knowledge graph (KG) embedding methods learn geometric representations of entities and relations to predict plausible missing knowledge. These representations are typically assumed to capture rule-like inference patterns. However, our theoretical understanding of the kinds of inference patterns that can be captured in this way remains limited. Ideally, KG embedding methods should be expressive enough such that for any set of rules, there exists an embedding that exactly captures these rules. This principle has been studied within the framework of region-based embeddings, but existing models are severely limited in the kinds of rule bases that can be captured. We argue that this stems from the use of representations that correspond to the Cartesian product of two-dimensional regions. As an alternative, we propose RESHUFFLE, a simple model based on ordering constraints that can faithfully capture a much larger class of rule bases than existing approaches. Moreover, the embeddings in our framework can be learned by a Graph Neural Network (GNN), which effectively acts as a differentiable rule base. This has some practical advantages, e.g. ensuring that embeddings can be easily updated as new knowledge is added to the KG. At the same time, since the resulting representations can be used similarly to standard KG embeddings, our approach is significantly more efficient than existing approaches to differentiable reasoning. The GNN-based formulation also allows us to study how bounded inference can be captured. We show in particular that bounded reasoning with arbitrary sets of closed path rules can be captured in this way.

## 1 Introduction

Knowledge graph (KG) embeddings (Bordes et al., 2013; Yang et al., 2015; Trouillon et al., 2016; Sun et al., 2019) are geometric representations of knowledge graphs. Such representations are typically used to infer plausible knowledge that is not explicitly stated in the KG. An important research question is concerned with the kinds of regularities that can be captured by different kinds of models. While standard approaches are often difficult to analyse from this perspective, region-based approaches make these regularities more explicit (Gutiérrez-Basulto & Schockaert, 2018; Abboud et al., 2020; Pavlovic & Sallinger, 2023; Charpenay & Schockaert, 2024). Essentially, in such approaches, each entity e is represented by an embedding $\mathbf{e} \in \mathbb{R}^d$ and each relation $r$ is represented by a geometric region $X_r \subseteq \mathbb{R}^{2d}$. We say that the triple $(e, r, f)$ is captured by the embedding iff $\mathbf{e} \oplus \mathbf{f} \in X_r$, where we write $\oplus$ for vector concatenation. In this way, we can naturally associate a KG with a given embedding. The key advantage of region-based models is that we can also associate a rule base with the embedding, where the rules reflect the spatial configuration of the regions $X_r$. However, not all rule bases can be captured in this way. As a simple example, models based on TransE (Bordes et al., 2013) cannot distinguish between the rules $r_1(X, Y) \wedge r_2(Y, Z) \to r_3(X, Z)$ and $r_2(X, Y) \wedge r_1(Y, Z) \to r_3(X, Z)$. This particular limitation can be avoided by using more sophisticated region-based models (Pavlovic & Sallinger, 2023; Charpenay & Schockaert, 2024), but even these models remain limited in terms of which rule bases they can capture. This appears to be related to the fact that these models use regions which are the Cartesian product of $d$ two-dimensional regions, i.e. $X_r = A_1^r \times ... \times A_d^r$, with $A_i^r \subseteq \mathbb{R}^2$. To check whether $(e, r, f)$ is captured, we then check whether $(e_i, f_i) \in A_i^r$ for each $i \in \{1, ..., d\}$, with $\mathbf{e} = (e_1, ..., e_d)$ and $\mathbf{f} = (f_1, ..., f_d)$. We will refer to such approaches as *coordinate-wise* models. Existing models primarily differ in how these two-dimensional regions are defined, e.g. ExpressivE (Pavlovic & Sallinger, 2023) uses

parallelograms for this purpose, while Charpenay & Schockaert (2024) used octagons. Using more flexible region-based representations typically leads to overfitting. In this paper, we go beyond coordinate-wise models but aim to avoid overfitting by otherwise keeping the model as simple as possible, by learning regions which are defined in terms of ordering constraints of the form $e_i \leq f_j$.

We show that this model, which we term RESHUFFLE, can capture a larger class of rule bases than existing region-based models. For instance, to the best of our knowledge, RESHUFFLE is the first that can capture (some) rule bases with cyclic dependencies. Furthermore, we show that entity embeddings in our framework can be learned using a monotonic Graph Neural Network (GNN) with randomly initialised node embeddings. This GNN effectively serves as a differentiable approximation of a rule base, acting on the initial representations of the entities to ensure that they capture the consequences that can be inferred from the KG. An important practical consequence is that our KG embeddings can be efficiently updated when new knowledge becomes available. Thus, our model is particularly well suited for KG completion in the inductive setting, where we need to predict links between entities that were not seen during training. From a theoretical point of view, the GNN-based formulation allows us to study bounded inference, where the number of layers of the GNN can be related to the number of inference steps. In particular, we show that our model is capable of faithfully capturing bounded inference with arbitrary sets of closed path rules. Finally, while the main focus of this paper is on advancing our theoretical understanding of the expressivity of knowledge graph embeddings, we also empirically evaluate RESHUFFLE on the task of inductive KG completion, where we find that it outperforms existing differentiable rule learning strategies.

## 2 RELATED WORK

**Region-based models** Despite the vast amount of work on KG embeddings in the last decade, the reasoning abilities of most existing models are poorly understood. The main exception comes from a line of work that has focused on region-based representations (Gutiérrez-Basulto & Schockaert, 2018; Abboud et al., 2020; Zhang et al., 2021; Leemhuis et al., 2022; Pavlovic & Sallinger, 2023; Charpenay & Schockaert, 2024). Essentially, the region-based view makes explicit which triples and rules are captured by a given embedding. This allows us to study what kinds of semantic dependencies a given model is capable of capturing, which is important for ensuring that models have the right inductive bias, especially for settings where reasoning is important. Existing work has uncovered various limitations of existing models. For instance, Gutiérrez-Basulto & Schockaert (2018) revealed that bilinear models such as RESCAL (Nickel et al., 2011), DistMult (Yang et al., 2015), TuckER (Balazevic et al., 2019) and ComplEx (Trouillon et al., 2016) cannot capture relation hierarchies in a faithful way. Gutiérrez-Basulto & Schockaert (2018) studied the expressivity of models where regions can be represented using arbitrary convex polytopes, finding that arbitrary sets of closed path rules can be faithfully captured by such representations. However, learning arbitrary polytopes is not feasible in practice for high-dimensional spaces, hence more recent works has focused on finding regions that are easier to learn while still retaining some of the theoretical advantages, such as Cartesian products of boxes (Abboud et al., 2020), cones (Zhang et al., 2021; Leemhuis et al., 2022), parallelograms (Pavlovic & Sallinger, 2023) and octagons (Charpenay & Schockaert, 2024). However, all these models are significantly more limited in the kinds of rules that they can capture. For instance, while the use of parallelograms and octagons makes it possible to capture some closed path rules, in practice we want to capture *sets* of such rules. This is only known to be possible under rather restrictive conditions (see Section 3).

**Inductive KG completion** Standard benchmarks for KG completion only test the reasoning abilities of models to a limited extent. For instance, BoxE (Abboud et al., 2020) achieves strong results on these benchmarks, despite provably being incapable of modelling simple rules such as $r_1(X,Y) \wedge r_2(Y,Z) \rightarrow r_3(X,Z)$. In our experiments, we will therefore focus on the problem of *inductive* KG completion (Teru et al., 2020). In the inductive setting, we need to predict links between entities that are different from those that were seen during training. To perform this task, models need to learn semantic dependencies between the relations, and then exploit this knowledge when making predictions. This can be achieved in different ways. A natural strategy is to learn rules from the training KG, either explicitly using models such as AnyBURL (Meilicke et al., 2019) and RNNLogic (Qu et al., 2021) or implicitly using differentiable rule learners such as Neural-LP (Yang et al., 2017), DRUM (Sadeghian et al., 2019) and NCRL (Cheng et al., 2023). In practice, better

results have been obtained using GNNs. For instance, some approaches (Teru et al., 2020) reduce the problem of link prediction to a graph classification problem. They first construct a subgraph containing paths connecting the head entity with some candidate tail entity, and then use a GNN to predict a score from this subgraph. Such approaches suffer from limited scalability, as answering a link prediction query requires constructing and processing such a subgraph for each candidate tail entity. NBFNet (Zhu et al., 2021) alleviates this limitation, by using a single GNN that processes the entire graph. The resulting node embeddings can then be used to score the different candidate tail entities. However, the node embeddings are query-specific, meaning that this model still requires a new forward pass of the GNN for each query, which is considerably less efficient than using KG embeddings. While we use a GNN for computing entity embeddings, once these embeddings have been learned, we can use them to answer arbitrary link prediction queries. RESHUFFLE is thus considerably more efficient than the aforementioned GNN-based models for inductive KG completion. ReFactor GNN (Chen et al., 2022) also uses a GNN to learn entity embeddings, by simulating the training dynamic of traditional KG embedding methods such as TransE (Bordes et al., 2013). However, their method has the disadvantage that all embeddings have to be recomputed when new triples are added to the KG. Moreover, their model inherits the limitations of traditional embedding models when it comes to faithfully modelling rules. Conceptually, our method has more in common with differentiable rule learning methods than with subgraph classification strategies. Indeed, each layer of the GNN updates the entity embeddings by essentially simulating the application of rules. Moreover, our model can simulate the deductive chaining of rules, which makes it fundamentally different from Neural-LP and DRUM, which focus on one-off rule application. Finally, while we focus on methods that learn by analysing the structure of a given KG, in some domains it is also possible to exploit prior knowledge, for instance by leveraging LLMs (Zhu et al., 2024).

## 3 PROBLEM SETTING

Let $\mathcal{R}$ be a set of relations, $\mathcal{E}$ a set of entities, and $\mathcal{G} \subseteq \mathcal{E} \times \mathcal{R} \times \mathcal{E}$ a knowledge graph. Similar to standard KG embedding models, our aim is to learn a vector $\mathbf{e} \in \mathbb{R}^d$ for every entity $e \in \mathcal{E}$ and a scoring function $s_r$ for every relation $r \in \mathcal{R}$ such that $s_r(\mathbf{e}, \mathbf{f})$ reflects the plausibility of the triple $(e, r, f)$. In region-based models, this scoring function is defined in terms of a geometric region $X_r \subseteq \mathbb{R}^d$. Specifically, the triple $(e, r, f)$ is then considered to be captured by the embedding iff $\mathbf{e} \oplus \mathbf{f} \in X_r$, where we write $\mathbf{e} \oplus \mathbf{f}$ to denote vector concatenation. The scoring function $s_r$ then reflects how close $\mathbf{e} \oplus \mathbf{f}$ is to the region $X_r$ (which is formalised in different ways by different models). A key advantage of region-based models is that they offer a mechanism for modelling rules. Let us write $\eta$ to denote a given region-based embedding, i.e. $\eta(e) \in \mathbb{R}^d$ denotes the embedding of the entity $e \in \mathcal{E}$ and $\eta(r) \subseteq \mathbb{R}^{2d}$ denotes the region representing the relation $r \in \mathcal{R}$. Similar to existing (differentiable) rule-based methods for KG completion (Yang et al., 2017; Meilicke et al., 2019; Sadeghian et al., 2019; Qu et al., 2021; Cheng et al., 2023), we focus on so-called *closed path rules*, which are rules $\rho$ of the following form:

$$r_1(X_1, X_2) \wedge r_2(X_2, X_3) \wedge ... \wedge r_p(X_p, X_{p+1}) \rightarrow r(X_1, X_{p+1}) \tag{1}$$

We refer to $r_1(X_1, X_2) \wedge r_2(X_2, X_3) \wedge ... \wedge r_p(X_p, X_{p+1})$ as the body of the rule and to $r(X_1, X_{p+1})$ as the head. We say that $\eta$ captures this rule if for all vectors $\mathbf{x_1}, ..., \mathbf{x_{p+1}} \in \mathbb{R}^n$ we have:

$$(\mathbf{x_1} \oplus \mathbf{x_2} \in \eta(r_1)) \wedge .... \wedge (\mathbf{x_p} \oplus \mathbf{x_{p+1}} \in \eta(r_p)) \Rightarrow (\mathbf{x_1} \oplus \mathbf{x_{p+1}} \in \eta(r)) \tag{2}$$

Apart from their practical significance, our focus on closed path rules is also motivated by the observation that existing region-based models have particular limitations when it comes to capturing this kind of rules. Some approaches, such as BoxE (Abboud et al., 2020) are not capable of capturing such rules at all. More recent approaches (Pavlovic & Sallinger, 2023; Charpenay & Schockaert, 2024) are capable of capturing closed path rules, but with significant limitations. Specifically, given a set of closed path rules $\mathcal{P}$, we ideally want an embedding $\eta$ that captures every rule in $\mathcal{P}$ while not capturing any rules that are not entailed by $\mathcal{P}$. Charpenay & Schockaert (2024) showed this to be possible, provided that every non-trivial rule entailed from $\mathcal{P}$ is a closed path rule in which $r_1, ..., r_p, r$ are all distinct relations. For instance, rules of the form $r_1(X_1, X_2) \wedge r_1(X_2, X_3) \rightarrow r(X_1, X_3)$ and $r_1(X_1, X_2) \wedge r_2(X_2, X_3) \rightarrow r_1(X_1, X_3)$ were not allowed in their construction. Similarly, they cannot capture rule bases with cyclic dependencies such as $\mathcal{P} = \{r_1(X_1, X_2) \wedge r_2(X_2, X_3) \rightarrow r_3(X_1, X_3), r_3(X_1, X_2) \wedge r_4(X_2, X_3) \rightarrow r_1(X_1, X_3)\}$.

In the following, we write $\mathcal{P} \cup \mathcal{G} \models (e, r, f)$ to denote that the triple $(e, r, f)$ can be entailed from the rule base $\mathcal{P}$ and the knowledge graph $\mathcal{G}$. More precisely, we have $\mathcal{P} \cup \mathcal{G} \models (e, r, f)$ iff either $(e, r, f) \in \mathcal{G}$ or $\mathcal{P}$ contains a rule of the form (1) such that $\mathcal{P} \cup \mathcal{G} \models (e, r_1, e_2)$, $\mathcal{P} \cup \mathcal{G} \models (e_2, r_2, e_3)$, ..., $\mathcal{P} \cup \mathcal{G} \models (e_p, r_p, f)$ for some entities $e_2, ..., e_p$. We furthermore write $\mathcal{P} \models \rho$ for a rule $\rho$ of the form (1) to denote that $\mathcal{P}$ entails $\rho$ w.r.t. the standard notion of entailment from propositional logic (when interpreting rules in terms of material implication). Note that while we consider both a knowledge graph $\mathcal{G}$ and a rule base $\mathcal{P}$ in our analysis, in practice only the knowledge graph $\mathcal{G}$ is given. We study whether our model is capable of capturing rule bases because this is a necessary condition to allow it to *learn* semantic dependencies in the form of rules.

## 4 MODELING RELATIONS USING ORDERING CONSTRAINTS

Our aim is to introduce a knowledge graph embedding model which is more general than existing coordinate-wise region-based embeddings, but which remains simple enough to make the representations learnable in practice. The central idea is to rely on ordering constraints. Specifically, we model each relation $r$ using a region $X_r$ of the following form:

$$X_r = \{(e_1, ..., e_d, f_1, ..., f_d) \,|\, \forall i \in I_r . e_{\sigma_r(i)} \leq f_i\} \qquad (3)$$

where the representation of a region $r$ is parameterised by a set of coordinates $I_r \subseteq \{1, ..., d\}$ and a mapping $\sigma_r : I_r \to \{1, ..., d\}$. We thus need a maximum of $2d$ parameters to completely specify the embedding of a given relation. Note that in the special case where $I_r = \emptyset$, we have $X_r = \mathbb{R}^d$.

**Example 1.** *Let $\mathbf{e} = (0, 0, 0)$, $\mathbf{f} = (0, 0, 1)$ and $\mathbf{g} = (2, 2, 0)$ be the embeddings of entities $e, f, g$. Let the relations $r_1, r_2, r_3$ be represented as follows: $I_{r_1} = \{3\}$, $I_{r_2} = \{1, 2\}$, $I_{r_3} = \{1\}$, $\sigma_{r_1}(3) = 2$, $\sigma_{r_2}(1) = \sigma_{r_2}(2) = 3$ and $\sigma_{r_3}(1) = 2$. Then we find that $\mathbf{e} \oplus \mathbf{f} \in X_{r_1}$, meaning that the triple $(e, r_1, f)$ is captured. Indeed, for $\mathbf{e} \oplus \mathbf{f} \in X_{r_1}$ to hold, we need $e_{\sigma_{r_1}(3)} = e_2 \leq f_3$, which is satisfied. We similarly find that $(f, r_2, g)$ is captured, because $f_{\sigma_{r_2}(1)} = f_3 \leq g_1$ and $f_{\sigma_{r_2}(2)} = f_3 \leq g_2$.*

The following example illustrates how the use of ordering constraints allows us to capture rules.

**Example 2.** *Consider a rule of the form $r_1(X, Y) \wedge r_2(Y, Z) \to r_3(X, Z)$. This rule is captured by an embedding of the form (3) if for each $i \in I_{r_3}$ we have that $i \in I_{r_2}$, $\sigma_{r_2}(i) \in I_{r_1}$ and $\sigma_{r_1}(\sigma_{r_2}(i)) = \sigma_{r_3}(i)$. For instance, the relations $r_1, r_2, r_3$ from Example 1 satisfy these conditions. In general, if these conditions are satisfied and we have $(e, r_1, f)$ and $(f, r_2, g)$ in $\mathcal{G}$, then for each $i \in I_{r_3}$ we have: $e_{\sigma_{r_1}(\sigma_{r_2}(i))} \leq f_{\sigma_{r_2}(i)} \leq g_i$. Since we assumed $\sigma_{r_1}(\sigma_{r_2}(i)) = \sigma_{r_3}(i)$ it follows that $e_{\sigma_{r_3}(i)} \leq g_i$ for every $i \in I_r$ and thus that the embedding captures the triple $(e, r_3, f)$.*

We will come back to the analysis of how rules can be modelled using ordering constraints in the next section. We now turn our focus to how (a differentiable approximation of) the ordering constraints can be learned. Note that we can characterise $X_r$ as follows:

$$X_r = \{\mathbf{e} \oplus \mathbf{f} \,|\, \max(\mathbf{A_r}\mathbf{e}, \mathbf{f}) = \mathbf{f}\} \qquad (4)$$

where the maximum is applied component-wise and the matrix $\mathbf{A_r} \in \mathbb{R}^{d \times d}$ is constrained such that (i) all components are either 0 or 1 and (ii) at most one component in each row is non-zero. The format of (4) suggests how entity embeddings in our framework can be learned using a GNN. A practical advantage of using a GNN for this purpose is that we can use our model for inductive KG completion. As we will see in the next section, the use of a GNN also has an important theoretical advantage, as it allows us to capture bounded reasoning with arbitrary sets of closed path rules.

**Learning embeddings with GNNs** Let us write $\mathbf{e}^{(l)} \in \mathbb{R}^d$ for the representation of entity $e$ in layer $l$ of the GNN. The embeddings $\mathbf{e}^{(0)}$ are initialised randomly, such that (i) all coordinates are non-negative, (ii) the coordinates of different entity embeddings are sampled independently, and (iii) there are at least two distinct values that have a non-negative probability of being sampled for each coordinate. Starting from (4), we naturally end up with the following message-passing GNN:

$$\mathbf{f}^{(l+1)} = \max\left(\{\mathbf{f}^{(l)}\} \cup \{\mathbf{A_r}\mathbf{e}^{(l)} \,|\, (e, r, f) \in \mathcal{G}\}\right) \qquad (5)$$

However, because the model relies on randomly initialised entity embeddings, the dimensionality of the entity embeddings needs to be sufficiently high. At the same time, the number of parameters that

have to be learned for each relation should be sufficiently low to prevent overfitting. For this reason, we decouple the number of parameters from the dimensionality of the embeddings. Specifically, we learn matrices $\mathbf{A_r}$ of the following form:

$$\mathbf{A_r} = \mathbf{B_r} \otimes \mathbf{I_k} \tag{6}$$

where we write $\otimes$ for the Kronecker product, $\mathbf{I_k}$ is the $k$-dimensional identity matrix and $\mathbf{B_r}$ is an $\ell \times \ell$ matrix, with $d = k\ell$, where the rows of $\mathbf{B_r}$ are constrained similarly as those of $\mathbf{A_r}$, i.e. each row is either a one-hot vector or a 0-vector. To make the computation of the GNN updates more efficient, we represent each entity using a matrix $\mathbf{Z_e^{(1)}} \in \mathbb{R}^{\ell \times k}$ and compute the updates as follows:

$$\mathbf{Z_f^{(l+1)}} = \max \left( \{\mathbf{Z_f^{(l)}}\} \cup \{\mathbf{B_r}\mathbf{Z_e^{(l)}} \mid (e, r, f) \in \mathcal{G}\} \right) \tag{7}$$

We will refer to this model as RESHUFFLE. Note that a triple $(e, r, f)$ is captured at layer $l$ if:

$$\mathbf{B_r}\mathbf{Z_e^{(l)}} \preceq \mathbf{Z_f^{(l)}}$$

where $\mathbf{X} \preceq \mathbf{Y}$ denotes that $\max(\mathbf{X}, \mathbf{Y}) = \mathbf{Y}$. A rule of the form (1) is satisfied if:

$$\mathbf{B_{r_p}}\mathbf{B_{r_{p-1}}} \cdots \mathbf{B_{r_1}} \preceq \mathbf{B_r} \tag{8}$$

In practice, we learn a soft approximation of the matrices $\mathbf{B_r}$. Specifically, to learn the matrix $\mathbf{B_r}$, we choose each row $i$ as the first $\ell$ coordinates of a vector $\mathsf{softmax}(b_{i,1}^r, ..., b_{i,\ell+1}^r)$, where $b_{i,1}^r, ..., b_{i,\ell+1}^r$ are learnable parameters. Note that we need $\ell+1$ parameters to allow for the possibility of some rows to be all 0s, which we empirically found to be important. The number of parameters per relation is thus quadratic in $\ell$. However, due to the use of the softmax operation, these representations can still be learned effectively (Lavoie et al., 2023). We experimented with a number of further strategies for imposing sparsity, but were not able to outperform the basic softmax formulation.

## 5 CONSTRUCTING GNNs FROM RULE GRAPHS

In equation (8) we already showed how a given closed path rule can be captured in RESHUFFLE. However, our main question of interest is whether it is possible to faithfully capture a *set* of closed path rules $\mathcal{P}$. More precisely, in this paper we study the following question: can parameters be found for the matrices $\mathbf{B_r}$ such that *all rules* entailed by $\mathcal{P}$ are captured, and *only those rules*. Rather than constructing the matrices $\mathbf{B_r}$ directly, we first introduce the notion of a rule graph, which will serve as a convenient abstraction for studying this problem. We then explain how we can construct the matrices $\mathbf{B_r}$ from a given rule graph. Throughout this paper, we will assume that $\mathcal{G}$ contains the triple $(e, eq, e)$ for every $e \in \mathcal{E}$, with $eq$ a relation which does not appear in the rule base $\mathcal{P}$. This assumption corresponds to the common practice of adding self-loops GNN models.

**Rule graphs** We associate with the rule base $\mathcal{P}$ a labelled multi-graph $\mathcal{H}$, i.e. a set of triples $(n_1, r, n_2)$. Note that this graph is formally equivalent to a knowledge graph, but the nodes in this case do not correspond to entities. Rather, as we will see, they correspond to the different rows/columns of the matrices $\mathbf{B_r}$. A path in $\mathcal{H}$ from $n_1$ to $n_{p+1}$ is a sequence of triples of the form $(n_1, r_1, n_2), (n_2, r_2, n_3), ..., (n_p, r_p, n_{p+1})$. The *type* of this path is given by the sequence of relations $r_1; r_2; ...; r_p$. The *eq-reduced type* of the path is obtained by removing all occurrences $eq$ in $r_1; r_2; ...; r_p$. For instance, for a path of type $r_1; eq; eq; r_2; eq$, the *eq*-reduced type is $r_1; r_2$.

**Definition 1.** *A rule graph $\mathcal{H}$ for a given rule base $\mathcal{P}$ is a labelled multi-graph, where the labels are taken from $\mathcal{R}$, such that the following properties are satisfied:*

**(R1)** *For every relation $r \in \mathcal{R}$, there is some edge in $\mathcal{H}$ labelled with $r$.*

**(R2)** *For every node $n$ in $\mathcal{H}$ and every $r \in \mathcal{R}$, it holds that $n$ has at most one incoming $r$-edge.*

**(R3)** *Suppose there is an $r$-edge in $\mathcal{H}$ from node $n_1$ to node $n_2$. Suppose furthermore that $\mathcal{P} \models r_1(X_1, X_2) \wedge r_2(X_2, X_3) \wedge ... \wedge r_p(X_p, X_{p+1}) \rightarrow r(X_1, X_{p+1})$. Then there is a path in $\mathcal{H}$ from $n_1$ to $n_2$ whose eq-reduced type is $r_1; ...; r_p$.*

**(R4)** *Suppose for every two nodes connected by an $r$-edge, there is a path connecting these nodes whose eq-reduced type belongs to $\{(r_{11}; ...; r_{1p_1}), ..., (r_{q1}; ...; r_{qp_q})\}$. Then there is some $i \in \{1, ..., q\}$ such that that $\mathcal{P} \models r_{i1}(X_1, X_2) \wedge ... \wedge r_{ip_i}(X_{p_i}, X_{p_{i+1}}) \rightarrow r(X_1, X_{p_{i+1}})$.*

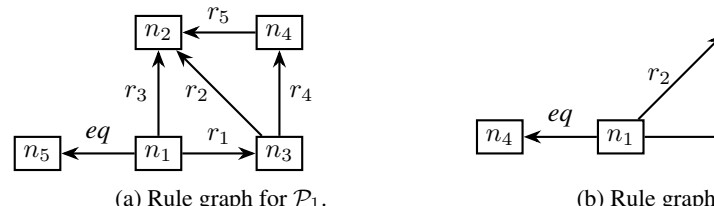

(a) Rule graph for $\mathcal{P}_1$.   (b) Rule graph for $\mathcal{P}_2$.

Figure 1: Rule graphs for the rule bases from Example 3.

This definition reflects the fact that a rule is captured when the ordering constraints associated with its body entail the ordering constraints associated with its head, as was illustrated in Example 2. Specifically, this requirement is captured by condition (R3). Condition (R4) is needed to ensure that only the rules in $\mathcal{P}$ are captured. Conditions (R1) and (R2) are needed because, in the construction we consider below, the nodes of the rule graph will correspond to the rows of the matrices $\mathbf{B_r}$. Condition (R1) ensures that $\mathbf{B_r}$ contains at least one non-zero component for each relation $r$, while (R2) ensures that each row of $\mathbf{B_r}$ has at most one non-zero component.

**Example 3.** *Let $\mathcal{P}_1$ contain the following rules:*

$$r_1(X,Y) \wedge r_2(Y,Z) \to r_3(X,Z)$$
$$r_4(X,Y) \wedge r_5(Y,Z) \to r_2(X,Z)$$

*A corresponding rule graph is shown in Figure 1a. Next, we illustrate how rule graphs can sometimes be constructed for rule bases with cyclic dependencies. Let $\mathcal{P}_2$ contain the following rules:*

$$r_2(X,Y) \wedge r_3(Y,Z) \to r_1(Y,Z)$$
$$r_1(X,Y) \wedge r_4(Y,Z) \to r_2(X,Z)$$

*A corresponding rule graph is shown in Figure 1b.*

**Constructing GNNs**  Given a rule graph $\mathcal{H}$, we define the corresponding parameters of the GNN as follows. Specifically, we need to define the matrix $\mathbf{B_r}$ for every $r$. Each node from the rule graph is associated with one row/column of $\mathbf{B_r}$. Let $n_1, ..., n_\ell$ be an enumeration of the nodes in the rule graph. The corresponding matrix $\mathbf{B_r} = (b_{ij})$ is defined as:

$$b_{ij} = \begin{cases} 1 & \text{if } \mathcal{H} \text{ has an } r\text{-edge from } n_j \text{ to } n_i \\ 0 & \text{otherwise} \end{cases} \tag{9}$$

Note that because of condition (R2), there will be at most one non-zero element in each row of $\mathbf{B_r}$, in accordance with the assumptions that we made in Section 4.

The following result shows that the constructed GNN indeed captures all the rules from $\mathcal{P}$. Specifically, we show that the embeddings which are learned by the GNN (upon convergence) capture all triples that are entailed by $\mathcal{P} \cup \mathcal{G}$. Note that, thanks to the use of the maximum in (7), the GNN always converges after a finite number of iterations.

**Proposition 1.** *Let $\mathcal{P}$ be a rule base and $\mathcal{G}$ a knowledge graph. Suppose $\mathcal{P} \cup \mathcal{G} \models (a, r, b)$. Let $\mathcal{H}$ be a rule graph for $\mathcal{P}$ and let $\mathbf{Z_e^{(1)}}$ be the entity representations that are learned by the corresponding* RESHUFFLE *model, as defined in (9). Assume $\mathbf{Z_e^{(m)}} = \mathbf{Z_e^{(m+1)}}$ for every entity $e$ $(m \in \mathbb{N})$. It holds that $\mathbf{B_r Z_a^{(m)}} \preceq \mathbf{Z_b^{(m)}}$.*

We also need to show that the GNN does not capture rules which are not entailed by $\mathcal{P}$. However, for any triple $(e, r, f)$ there is always a chance that it is captured by the model, even if $\mathcal{P} \cup \mathcal{G} \not\models (e, r, f)$, due to the fact that the entity embeddings are initialised randomly. However, by choosing the dimensionality of the entity embeddings to be sufficiently large, we can make the probability of this happening arbitrarily small. As before, we write $\ell$ to denote the number of rows in $\mathbf{Z}_e$ and $k$ for the number of columns. Note that the value of $k$ does not affect the number of parameters of the model, since the size of the matrices $\mathbf{B_r}$ only depends on $\ell$ and the entity embeddings are randomly initialised. In practice, we can thus simply choose $k$ to be sufficiently large.

**Proposition 2.** *Let $\mathcal{P}$ be a rule base and $\mathcal{G}$ a knowledge graph. Let $\mathcal{H}$ be a rule graph for $\mathcal{P}$ and let $\mathbf{Z}_{\mathbf{e}}^{(1)}$ be the entity representations that are learned by the corresponding* RESHUFFLE *model, as defined in (9). For any $\varepsilon > 0$, there exists some $k_0 \in \mathbb{N}$ such that, when $k \geq k_0$, for any $m \in \mathbb{N}$ and $(a, r, b) \in \mathcal{E} \times \mathcal{R} \times \mathcal{E}$ such that $\mathcal{P} \cup \mathcal{G} \not\models (a, r, b)$, we have*

$$Pr[\mathbf{B_r Z_a^{(m)}} \preceq \mathbf{Z_b^{(m)}}] \leq \varepsilon$$

## 6 Constructing rule graphs

We now return to the central question of this paper: given a rule base $\mathcal{P}$, is it possible to construct a RESHUFFLE model which captures the rules entailed by $\mathcal{P}$ and only those rules? Thanks to Propositions 1 and 2 we know that this is the case when a rule graph for $\mathcal{P}$ exists. The key question thus becomes whether it is always possible to construct such a rule graph. As the following result shows, if there are no cyclic dependencies in $\mathcal{P}$, a rule graph always exits.

**Proposition 3.** *Let $\mathcal{P}$ be a rule base. Assume that we can rank the relations in $\mathcal{R}$ as $r_1, ..., r_{|\mathcal{R}|}$, such that for every rule in $\mathcal{P}$ with $r_i$ in the body and $r_j$ in the head, it holds that $i < j$. There exists a rule graph for $\mathcal{P}$.*

It follows in particular that the class of rule bases that can be captured with RESHUFFLE models is strictly larger than the class of rule bases that has been considered in previous work (Charpenay & Schockaert, 2024). Unfortunately, it turns out that there exist rule bases with cyclic dependencies for which no valid rule graph can be found. This is illustrated in the next example.

**Example 4.** *Let $\mathcal{P}$ contain the following rule:*

$$r_1(X, Y) \wedge r_2(Y, Z) \wedge r_1(Z, U) \rightarrow r_2(X, U)$$

*To see why there is no rule graph for $\mathcal{P}$, consider the following knowledge graph $\mathcal{G}$:*

$$\mathcal{G} = \{(x_1, r_1, x_2), (x_2, r_1, x_3), ..., (x_{l-1}, r_1, r_l), (x_l, r_2, x_{l+1}), (x_{l+1}, r_1, x_{l+2}), ..., (x_k, r_1, x_{k+1})\}$$

*We have that $\mathcal{P} \cup \mathcal{G} \models (x_1, r_2, x_{k+1})$ only if the number of repetitions of $r_1$ at the start of the sequence matches the number of repetitions at the end, but rule graphs cannot encode this.*

The argument from the previous example can be formalised as follows. Let $\mathcal{P}$ be a set of closed path rules. Let $\mathcal{R}_1$ be the set of relations from $\mathcal{R}$ that appear in the head of some rule in $\mathcal{P}$. For any $r \in \mathcal{R}_1$, we can consider a context-free grammar with two types of production rules:

- For each rule of the form (1), there is a production rule $r \Rightarrow r_1 r_2 ... r_p$.
- For each $r \in \mathcal{R}_1$, there is a production rule $r \Rightarrow \overline{r}$.

The elements of $(\mathcal{R} \setminus \mathcal{R}_1) \cup \{\overline{r} \mid r \in \mathcal{R}_1\}$ are treated as terminal symbols, those in $\mathcal{R}_1$ as non-terminal symbols, and $r$ is the starting symbol. Let us write $L_r$ for the corresponding language.

**Proposition 4.** *Let $\mathcal{P}$ be a set of closed path rules and suppose that there exists a rule graph $\mathcal{H}$ for $\mathcal{P}$. Let $\mathcal{R}_1$ be the set of relations that appear in the head of some rule in $\mathcal{P}$. It holds that the language $L_r$ is regular for every $r \in \mathcal{R}_1$.*

This result shows that we cannot capture arbitrary rule bases using rule graphs. For instance, for the rule base from Example 4, we have $L_{r_2} = \{r_1^{(l)} \overline{r}_2 r_1^{(l)} \mid l \in \mathbb{N} \setminus \{0\}\}$, where we write $x^{(l)}$ for the string that consists of $l$ repetitions of $x$. It is well-known that the language $L_{r_2}$ is not regular, hence it follows from Proposition 4 that no rule graph exists for this rule base.

Following the negative result that arises from Proposition 4, we now establish two important positive results. First, in Section 6.1, inspired by regular grammars, we introduce a special class of rule bases with cyclic dependencies for which a rule graph is guaranteed to exist. Second, in Section 6.2, we focus on the practically important setting of bounded inference: since GNNs use a fixed number of layers in practice, what mostly matters is what can be derived in a bounded number of steps. It turns out that if we only care about such inferences, we can capture arbitrary sets of closed path rules.

## 6.1 LEFT-REGULAR RULE BASES

To show that many rule bases with cyclic dependencies can still be faithfully modelled, we consider the following notion of a left-regular rule base, inspired by left-regular grammars.

**Definition 2.** *Let $\mathcal{P}$ be a rule base. Let $\mathcal{R}_1$ be the set of relations that appear in the head of a rule from $\mathcal{P}$. We call $\mathcal{P}$ left-regular if every rule is of the following form:*

$$r_1(X, Y) \wedge r_2(Y, Z) \rightarrow r_3(X, Z) \tag{10}$$

*such that $r_2 \notin \mathcal{R}_1$.*

While Definition 2 only considers rules with two relations in the body, rules with more than two atoms can straightforwardly be simulated by introducing fresh relations. The following result shows that left-regular rule bases can always be faithfully captured by a RESHUFFLE model.

**Proposition 5.** *For any left-regular set of closed path rules $\mathcal{P}$, there exists a rule graph for $\mathcal{P}$.*

## 6.2 BOUNDED INFERENCE

In practice, the GNN can only carry out a finite number of inference steps. Rather than requiring that the resulting embeddings capture all triples that can be inferred from $\mathcal{P} \cup \mathcal{G}$, it is natural to merely require that the result captures all triples that can be inferred using a bounded number of inference steps. We know from Proposition 4 that it is not always possible to construct a rule graph for a given rule base $\mathcal{P}$. To address this, we now weaken the notion of a rule graph, aiming to capture reasoning up to a fixed number of inference steps. In the following, we will assume that $\mathcal{P}$ only contains rules with two relations in the body, i.e. rules such as the one in (4) (but without imposing the requirement that $r_2 \notin \mathcal{R}_1$). Note that we can assume this w.l.o.g. as any set of closed path rules can be converted in such a format by introducing fresh relations.

Let us write $\mathcal{P} \cup \mathcal{G} \models_m (e, r, f)$ to denote that $(e, r, f)$ can be derived from $\mathcal{P} \cup \mathcal{G}$ in $m$ steps. More precisely, we have $\mathcal{P} \cup \mathcal{G} \models_0 (e, r, f)$ iff $(e, r, f) \in \mathcal{G}$. Furthermore, we have $\mathcal{P} \cup \mathcal{G} \models_m (e, r, f)$, for $m > 0$, iff $\mathcal{P} \cup \mathcal{G} \models_{m-1} (e, r, f)$ or there is a rule $r_1(X_1, X_2) \wedge r_2(X_2, X_3) \rightarrow r(X_1, X_3)$ in $\mathcal{P}$ and an entity $g \in \mathcal{E}$ such that $\mathcal{P} \cup \mathcal{G} \models_{m_1} (e, r_1, g)$ and $\mathcal{P} \cup \mathcal{G} \models_{m_2} (g, r_2, f)$, with $m = m_1 + m_2 + 1$.

**Definition 3.** *Let $m \in \mathbb{N}$. We call $\mathcal{H}$ an $m$-bounded rule graph for $\mathcal{P}$ if $\mathcal{H}$ satisfies conditions (R1)–(R3) as well as the following weakening of (R4):*

**(R4m)** *Suppose for every two nodes connected by an $r$-edge, there is a path connecting these two nodes whose eq-reduced type belongs to $\{(r_{11}; ...; r_{1p_1}), ..., (r_{q1}; ...; r_{qp_q})\}$, with $p_1, ..., p_q \leq m + 1$. Then there is some $i \in \{1, ..., q\}$ such that that $\mathcal{P} \models_m r_{i1}(X_1, X_2) \wedge ... \wedge r_{ip_i}(X_{p_i}, X_{p_{i+1}}) \rightarrow r(X_1, X_{p_{i+1}})$.*

Given an $m$-bounded rule graph, we can construct a corresponding GNN in the same way as in Section 5. Moreover, Proposition 1 remains valid for $m$-bounded rule graphs, as its proof does not depend on (R4). Proposition 2 can be weakened as follows.

**Proposition 6.** *Let $\mathcal{P}$ be a rule base and $\mathcal{G}$ a knowledge graph. Let $\mathcal{H}$ be an $m$-bounded rule graph for $\mathcal{P}$ and let $\mathbf{Z}_{\mathbf{e}}^{(1)}$ be the entity representations that are learned by the corresponding RESHUFFLE model, as defined in (9). For any $\varepsilon > 0$, there exists some $k_0 \in \mathbb{N}$ such that, when $k \geq k_0$, for any $i \leq m + 1$ and $(a, r, b) \in \mathcal{E} \times \mathcal{R} \times \mathcal{E}$ such that $\mathcal{P} \cup \mathcal{G} \not\models_m (a, r, b)$, we have*

$$Pr[\mathbf{B_r Z_a^{(i)}} \preceq \mathbf{Z_b^{(i)}}] \leq \varepsilon$$

**Proposition 7.** *For any set of closed path rules $\mathcal{P}$, there exists an $m$-bounded rule graph for $\mathcal{P}$.*

## 7 EXPERIMENTAL RESULTS

Thus far, we have shown that RESHUFFLE is capable of capturing bounded reasoning for arbitrary sets of closed path rules, as well as complete reasoning for several important special cases. We now complement this theoretical analysis with an empirical evaluation, to show that suitable model parameters can be effectively learned in practice, and to compare the performance of RESHUFFLE with existing differentiable rule learning strategies. For this evaluation, we focus on the task of inductive KG completion, as the need to capture reasoning patterns is intuitively more important for this setting compared to the traditional (i.e. transductive) setting.

Table 1: Hits@10 for 50 negative samples on inductive KGC split by method type (GNN-based vs. rule-based vs. differentiable rule-based).

| | | FB15k-237 | | | | WN18RR | | | | NELL-995 | | | |
|---|---|---|---|---|---|---|---|---|---|---|---|---|---|
| | | v1 | v2 | v3 | v4 | v1 | v2 | v3 | v4 | v1 | v2 | v3 | v4 |
| GNN | CoMPILE | 0.676 | 0.829 | 0.846 | 0.874 | 0.836 | 0.798 | 0.606 | 0.754 | 0.583 | 0.938 | 0.927 | 0.751 |
| | GraIL | 0.642 | 0.818 | 0.828 | 0.893 | 0.825 | 0.787 | 0.584 | 0.734 | 0.595 | 0.933 | 0.914 | 0.732 |
| | NBFNet | 0.845 | 0.949 | 0.946 | 0.947 | 0.946 | 0.897 | 0.904 | 0.889 | 0.644 | 0.953 | 0.967 | 0.928 |
| Rule | RuleN | 0.498 | 0.778 | 0.877 | 0.856 | 0.809 | 0.782 | 0.534 | 0.716 | 0.535 | 0.818 | 0.773 | 0.614 |
| | AnyBURL | 0.604 | 0.823 | 0.847 | 0.849 | 0.867 | 0.828 | 0.656 | 0.796 | 0.683 | 0.835 | 0.798 | 0.652 |
| Diff-R | DRUM | 0.529 | 0.587 | 0.529 | 0.559 | 0.744 | 0.689 | 0.462 | 0.671 | 0.194 | 0.786 | 0.827 | 0.806 |
| | Neural-LP | 0.529 | 0.589 | 0.529 | 0.559 | 0.744 | 0.689 | 0.462 | 0.671 | 0.408 | 0.787 | 0.827 | 0.806 |
| | RESHUFFLE | 0.747 | 0.885 | 0.903 | 0.918 | 0.710 | 0.729 | 0.602 | 0.694 | 0.638 | 0.861 | 0.882 | 0.812 |

Table 2: Hits@10 for 50 negative samples on inductive KGC for each ablation of RESHUFFLE.

| | FB15k-237 | | | | WN18RR | | | | NELL-995 | | | |
|---|---|---|---|---|---|---|---|---|---|---|---|---|
| | v1 | v2 | v3 | v4 | v1 | v2 | v3 | v4 | v1 | v2 | v3 | v4 |
| RESHUFFLE$^2$ | 0.304 | 0.569 | 0.385 | 0.916 | 0.293 | 0.309 | 0.155 | 0.270 | 0.488 | 0.558 | 0.334 | 0.370 |
| RESHUFFLE$_{nL}$ | 0.744 | 0.890 | 0.903 | 0.917 | 0.698 | 0.685 | 0.618 | 0.682 | 0.627 | 0.738 | 0.886 | 0.815 |
| RESHUFFLE | 0.747 | 0.885 | 0.903 | 0.918 | 0.710 | 0.729 | 0.602 | 0.694 | 0.638 | 0.861 | 0.882 | 0.812 |

**Datasets**   We evaluate RESHUFFLE on the three standard benchmarks for inductive knowledge graph completion (KGC) that were derived by Teru et al. (2020) from three datasets: FB15k-237, WN18RR, and NELL-995. Each of these inductive benchmarks contains four different dataset variants, named v1 to v4, and each of these variants consists of two graphs, a training and a testing graph, which are sampled from the original dataset as follows. The *training graph* $\mathcal{G}_{Train}$ was obtained by randomly sampling different numbers of entities and selecting their $k$-hop neighbourhoods. Next, to construct a disjoint *testing graph* $\mathcal{G}_{Test}$, the entities of $\mathcal{G}_{Train}$ were removed from the initial graph, and the same sampling procedure was repeated. Each of these graphs was split into a train set (80%), validation set (10%), and test set (10%). Thus, the three inductive benchmarks consist in total of twelve datasets: FB15k-237 v1-4, WN18RR v1-4, and NELL-995 v1-4. Furthermore, each of these datasets consists of six graphs: the train, validation, and test splits of $\mathcal{G}_{Train}$ and $\mathcal{G}_{Test}$. The supplementary materials provide additional information about these benchmarks.

**Experimental setup**   Following Teru et al. (2020), we train RESHUFFLE on the train split of $\mathcal{G}_{Train}$, tune our model's hyper-parameters on the validation split of $\mathcal{G}_{Train}$, and finally evaluate the best model on the test split of $\mathcal{G}_{Test}$. As discussed by Anil et al. (2024), some approaches in the literature have been evaluated in different ways, e.g. by tuning hyper-parameters on the validation split of $\mathcal{G}_{Test}$, and their reported results are thus not directly comparable. To account for small performance fluctuations, we repeat our experiments three times and report RESHUFFLE's average performance.[1] For the final evaluation, we select the hyper-parameter configuration with the highest Hits@10 score on the validation split of $\mathcal{G}_{Train}$. In accordance with Teru et al. (2020), we evaluate RESHUFFLE's test performance on 50 negatively sampled entities per triple of the test split of $\mathcal{G}_{Test}$ and report the Hits@10 scores. We list further details about the experimental setup in the supplementary materials. To facilitate RESHUFFLE's reuse by our community, we will provide its source code in a public GitHub repository upon acceptance of our paper.

**Baselines**   As the analysis in Sections 5 and 6 reveals, our GNN model acts as a kind of differentiable rule base. We therefore compare RESHUFFLE to existing approaches for differentiable rule learning: Neural-LP (Yang et al., 2017) and DRUM (Sadeghian et al., 2019). We also compare our method to two classical rule learning methods: RuleN (Meilicke et al., 2018) and AnyBURL (Meilicke et al., 2019). Finally, we include a comparison with GNN-based approaches: CoMPILE (Mai et al., 2021), GraIL (Teru et al., 2020), and NBFNet (Zhu et al., 2021).

---

[1]Results for all seeds and the resulting standard deviations are provided in the supplementary materials.

**Results**    Table 1 reports the performance of RESHUFFLE on the inductive benchmarks. The results of RESHUFFLE were obtained by us; AnyBURL and NBFNet results are from Anil et al. (2024); Neural-LP, DRUM, RuleN, and GraIL results are from Teru et al. (2020); and CoMPILE results are from Mai et al. (2021). Table 1 reveals that RESHUFFLE consistently outperforms the differentiable rule learners DRUM and Neural-LP, often by a significant margin (with WN18RR-v1 the only exception). Compared to the traditional rule learners, RESHUFFLE performs clearly better on FB15k-237 and NELL-995 (apart from v1) but underperforms on the WN18RR benchmarks. Anil et al. (2024) found that the kind of rules which are needed for WN18RR are much noisier compared to those than those which are needed for FB15k-237 and NELL-995. Our use of ordering constraints may be less suitable in such cases. Finally, compared to the GNN-based methods, RESHUFFLE outperforms CoMPILE and GraIL on FB15k-237 and NELL-995 v1 and v4 while again (mostly) underperforming on WN18RR. RESHUFFLE consistently underperforms the state-of-the-art method NBFNet. Recall, however, that our approach is significantly more efficient than such GNN-based approaches, as RESHUFFLE can score the plausibility of a given triple almost instantaneously. In contrast, NBFNet (Zhu et al., 2021) requires one forward pass of the GNN for every query, whereas methods such as GraIL (Teru et al., 2020) even need one forward pass for each candidate link for every query. Moreover, thanks to the use of max-pooling in the GNN, our embeddings can straightforwardly be updated when new knowledge becomes available. Finally, as the analysis by Anil et al. (2024) revealed, the performance of rule based methods can be significantly improved by combining them with other methods. The main issue is that for many queries, no strong evidence is available for any of the answer candidates, which rule based methods struggle with. To outperform methods such as NBFNet, rule based approach thus need to be combined with some kind of fallback model. A detailed analysis of this is outside the scope of this work.

Finally, we empirically investigate RESHUFFLE's components. We consider two variants for this ablation study, namely: $(i)$ RESHUFFLE$_{nL}$, which does not add a self-loop relation to the KG (i.e. triples of the form $(e, eq, e)$); and $(ii)$ RESHUFFLE$^2$, which allows for more general $\mathbf{B_r}$ matrices. In particular, different from RESHUFFLE, which applies the softmax function on the rows of $\mathbf{B_r}$ (see Section 4), RESHUFFLE$^2$ squares the $\mathbf{B_r}$ matrices component-wise, thereby allowing them to contain arbitrary positive values. For a fair comparison, we train each of RESHUFFLE's versions with the same hyper-parameter values, experimental setup, and evaluation protocol (see supplementary materials). Table 2 depicts the outcome of this study. It reveals that RESHUFFLE performs comparable to or better than RESHUFFLE$_{nL}$ and dramatically outperforms RESHUFFLE$^2$ on all benchmarks. The similar performance of RESHUFFLE and RESHUFFLE$_{nL}$ on most datasets suggests that the self-loop relation only matters in specific cases, which may not occur frequently in some datasets. The poor performance of RESHUFFLE$^2$ is as expected since allowing arbitrary positive parameters makes overfitting the training data more likely.

## 8  CONCLUSIONS

The region-based view of KG embeddings makes it possible to formally analyse which kinds of inference patterns are captured by a given embedding. An important question, which was left unanswered by previous work, is whether a region-based embedding model can be found which is capable of capturing arbitrary sets of closed path rules, while still ensuring that embeddings can be learned effectively in practice. In this context, we proposed a novel approach based on ordering constraints between reshuffled entity embeddings. This model, called RESHUFFLE, was chosen because it allows us to escape the limitations of coordinate-wise approaches while otherwise remaining as simple as possible. We found that RESHUFFLE has several interesting properties. Most significantly, we showed that bounded reasoning with arbitrary sets of closed path rules can be faithfully captured. We also revealed two special cases where exact reasoning is possible, which go significantly beyond what is (known to be) possible with existing region based models. From a practical point of view, our GNN formulation enables an efficient approach to inductive KG completion, where the resulting entity embeddings can moreover be efficiently updated as new knowledge is added to the KG. Empirically, we found our approach to outperform existing differentiable rule learners, while underperforming the state-of-the-art more generally. This latter result reflects the fact that (differentiable) rule based methods are less suitable when we need to weigh different pieces of weak evidence. In such cases, when further evidence becomes available, we may want to revise earlier assumptions, which is not possible with RESHUFFLE. Developing effective models that can provably simulate non-monotonic (or probabilistic) reasoning thus remains as an important challenge for future work.

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

## A  CONSTRUCTING GNNs FROM RULE GRAPHS

Let $\mathcal{P}$ be a set of closed path rules and let $\mathcal{H}$ be a corresponding rule graph, satisfying the conditions (R1)–(R4). We also assume that a knowledge graph $\mathcal{G}$ is given. We show that the GNN, which is constructed based on $\mathcal{H}$, correctly simulates the rules from $\mathcal{P}$. For the proofs, it will be more convenient to characterise the GNN in terms of operations on the coordinates of entity embeddings. Specifically, let $Z_i = \{(i-1)k+1, ..., (i-1)k+k\}$ and let $N_r \subseteq \{n_1, ..., n_\ell\}$ be the set of nodes from the rule graph $\mathcal{H}$ which have an incoming edge labelled with $r$. We define:

$$I_r = \bigcup_{n_i \in N_r} Z_i$$

Let $n_i \in N_r$ and let $(n_j, n_i)$ be the unique incoming edge with label $r$. Then we define ($t \in \{1, ..., k\}$):

$$\sigma_r((i-1)k+t) = (j-1)k+t$$

Now let us define:

$$\mu_r(e_1, ..., e_d) = (e'_1, ..., e'_d)$$

where $e'_i = e_{\sigma_r(i)}$ if $i \in I_r$ and $e'_i = 0$ otherwise. Let $\mathbf{e}^{(l)}$ be the entity embedding corresponding to the matrix $\mathbf{Z_e^{(1)}}$. In other words, if we write $z_{ij}$ for the components of $\mathbf{Z_e^{(1)}}$ and $e_i$ for the components of $\mathbf{e}^{(l)}$, then we have $z_{ij} = e_{(i-1)k+j}$. For a matrix $\mathbf{X} = (x_{ij})$, let us write *flatten*$(\mathbf{X})$ for the vector that is obtained by concatenating the rows of $\mathbf{X}$. In particular, *flatten*$(\mathbf{Z_e^{(1)}}) = \mathbf{e}^{(l)}$. The following lemma reveals how the GNN constructed from the rule graph $\mathcal{H}$ can be characterised in terms of entity embeddings.

**Lemma 1.** *It holds that flatten*$(\mathbf{B_r Z_e^{(1)}}) = \mu_r(\mathbf{e}^{(l)})$.

*Proof.* Let us write *flatten*$(\mathbf{B_r Z_e^{(1)}}) = (x_1, ..., x_d)$, $\mu_r(\mathbf{e}^{(l)}) = (y_1, ..., y_d)$ and $\mathbf{e}^{(l)} = (e_1, ..., e_d)$. Let $i \in \{1, ..., \ell\}$. Let us first assume that $n_i$ does not have any incoming edges in $\mathcal{H}$ which are labelled with $r$. In that case, row $i$ of $\mathbf{B_r}$ consists only of 0s and we have $x_{(i-1)k+1} = ... = x_{(i-1)k+k} = 0$. Similarly, we then also have $(i-1)k+j \notin I_r$ for $j \in \{1, ..., k\}$ and thus $y_{(i-1)k+1} = ... = y_{(i-1)k+k} = 0$. Now assume that there is an edge from $n_j$ to $n_i$ which is labelled with $r$. Then we have that row $i$ of $\mathbf{B_r}$ is a one-hot vector with 1 at position $j$. Accordingly, we have $x_{(i-1)k+t} = e_{(j-1)k+t}$ for $t \in \{1, ..., k\}$. Accordingly we then have $\sigma_r((i-1)k+t) = (j-1)k+t$ and thus $y_{(i-1)k+t} = e_{(j-1)k+t}$. $\qquad\square$

For a sequence of relations $r_1, ..., r_p$, we define $\mu_{r_1;...;r_p}$ as follows. We define $\mu_{r_1;...;r_p}(x_1, ..., x_d) = (y_1, ..., y_d)$, where ($i \in \{1, ..., \ell\}$, $t \in \{1, ..., k\}$):

$$y_{(i-1)k+t} = \begin{cases} x_{(j-1)k+t} & \text{if there is an } r_1; ...; r_p \text{ path} \\ & \text{from } n_j \text{ to } n_i \\ 0 & \text{otherwise} \end{cases}$$

Note that if there is an $r_1; ...; r_k$ path arriving at node $n_i$ in the rule graph, it has to be unique, given that each node has at most one incoming edge of a given type. In the following, we will also use $I_{r_1;...;r_p}$, defined as follows:

$$I_{r_1;...;r_p}$$
$$= \{(i-1)k + t \,|\, \text{there is an } r_1; ...; r_p \text{ path ending in } n_i\}$$

We have the following result.

**Lemma 2.** *For $r_1, ..., r_p \in \mathcal{R}$ we have*

$$\mu_{r_1;...;r_p}(x_1, ..., x_d) = \mu_{r_p}(...\mu_{r_1}(x_1, ..., x_d)...)$$

*Proof.* It is sufficient to show

$$\mu_{r_1;...;r_p}(x_1, ..., x_d) = \mu_{r_p}(\mu_{r_1;...;r_{p-1}}(x_1, ..., x_d))$$

We have $\mu_{r_1;...;r_{p-1}}(x_1, ..., x_d) = (y_1, ..., y_d)$, with

$$y_{(i-1)k+t} = \begin{cases} x_{(j-1)k+t} & \text{if there is an } r_1; ...; r_{p-1} \text{ path} \\ & \text{from } n_j \text{ to } n_i \\ 0 & \text{otherwise} \end{cases}$$

We furthermore have $\mu_{r_p}(y_1, ..., y_d) = (z_1, ..., z_d)$ with

$$z_{(i-1)k+t} = \begin{cases} y_{(j-1)k+t} & \text{if there is an } r_p\text{-edge} \\ & \text{from } n_j \text{ to } n_i \\ 0 & \text{otherwise} \end{cases}$$

Taking into account the definition of $(y_1, ..., y_d)$, we have $y_{(j-1)k+t} \neq 0$ only if there is an $r_1; ...; r_{p-1}$ path from some node $n_l$ to the node $n_j$, in which case we have $y_{(j-1)k+t} = x_{(l-1)k+t}$. In other words, we have:

$$z_{(i-1)k+t} = \begin{cases} x_{(l-1)k+t} & \text{if there is an } r_1; ...; r_{p-1} \text{ path} \\ & \text{from } n_l \text{ to some } n_j \text{ and an} \\ & r_p \text{ edge from } n_j \text{ to } n_i \\ 0 & \text{otherwise} \end{cases}$$

In other words, we have

$$z_{(i-1)k+t} = \begin{cases} x_{(l-1)k+t} & \text{if there is an } r_1; ...; r_p \text{ path} \\ & \text{from } n_l \text{ to } n_i \\ 0 & \text{otherwise} \end{cases}$$

We thus have $(z_1, ..., z_d) = \mu_{r_1;...;r_p}(x_1, ..., x_d)$. $\square$

We also have the following result.

**Lemma 3.** *Suppose $\mathcal{P} \models r_1(X_1, X_2) \wedge r_2(X_2, X_3) \wedge ... \wedge r_p(X_p, X_{p+1}) \rightarrow r(X_1, X_{p+1})$. There exists paths of type $r_1^1; ...; r_{q_1}^1$ and $r_1^2; ...; r_{q_2}^2$ and ... and $r_1^l; ...; r_{q_l}^l$, all of whose eq-reduced type is $r_1; ...; r_p$, such that for every embedding $(x_1, ..., x_d)$ we have:*

$$\mu_r(x_1, ..., x_d) \preccurlyeq \max_{i=1}^{l} \mu_{r_1^i;...;r_{q_i}^i}(x_1, ..., x_d)$$

*Proof.* This follows immediately from the fact that whenever there is an $r$-edge between two nodes $n$ and $n'$, there must also be a path between these nodes whose *eq*-reduced type is $r_1; ...; r_p$, because of condition (R3). $\square$

The following result shows that the GNN will correctly predict all triples that can be inferred from $\mathcal{G} \cup \mathcal{P}$.

**Proposition 8.** *Let $\mathcal{P}$ be a rule base and $\mathcal{G}$ a knowledge graph. Suppose $\mathcal{P} \cup \mathcal{G} \models (a, r, b)$. Let $\mathcal{H}$ be a rule graph for $\mathcal{P}$ and let $\mathbf{Z}_{\mathbf{e}}^{(\mathbf{1})}$ be the entity representations that are learned by the corresponding GNN. Assume $\mathbf{Z}_{\mathbf{e}}^{(\mathbf{m})} = \mathbf{Z}_{\mathbf{e}}^{(\mathbf{m+1})}$ for every entity $e$ ($m \in \mathbb{N}$). It holds that $\mathbf{B_r} \mathbf{Z}_{\mathbf{a}}^{(\mathbf{m})} \preceq \mathbf{Z}_{\mathbf{b}}^{(\mathbf{m})}$.*

*Proof.* Because of Lemma 1, it is sufficient to show that $\mu_r(\mathbf{a^{(m)}}) \preceq \mathbf{b^{(m)}}$. If $\mathcal{G}$ contains the triple $(a, r, b)$ then the result is trivially satisfied. Otherwise, $\mathcal{P} \cup \mathcal{G} \models r(a, b)$ implies that $\mathcal{P} \models r_1(X_1, X_2) \wedge r_2(X_2, X_3) \wedge ... \wedge r_p(X_p, X_{p+1}) \rightarrow r(X_1, X_{p+1})$, for some $r_1, ..., r_p, r \in \mathcal{R}$ such that $\mathcal{G}$ contains triples $(a, r_1, a_2), (a_2, r_2, a_3), ..., (a_p, r_p, b)$, for some $a_2, ..., a_p \in \mathcal{E}$. Because $(a, r_1, a_2) \in \mathcal{G}$, by construction, it holds for each $i \in \mathbb{N}$ that:

$$\mu_{r_1}(\mathbf{a^{(i)}}) \preccurlyeq \mathbf{a_2^{(i+1)}}$$

Similarly, because $(a_2, r_2, a_3) \in \mathcal{G}$, we have $\mu_{r_2}(\mathbf{a_2^{(i+1)}}) \preccurlyeq \mathbf{a_3^{(i+2)}}$ and thus

$$\mu_{r_2}(\mu_{r_1}(\mathbf{a^{(i)}})) \preccurlyeq \mu_{r_2}(\mathbf{a_2^{(i+1)}}) \preccurlyeq \mathbf{a_3^{(i+2)}}$$

In other words, we have

$$\mu_{r_1; r_2}(\mathbf{a^{(i)}}) \preccurlyeq \mathbf{a_3^{(i+2)}}$$

Continuing in the same way, we find that

$$\mu_{r_1; ...; r_{p-1}; r_p}(\mathbf{a^{(i)}}) \preccurlyeq \mathbf{b^{(i+p)}}$$

Now consider a path of type $r_1'; ...; r_q'$ whose *eq*-reduced type is $r_1; ...; r_p$. Then we have that $\mathcal{G}$ contains triples of the form $(a, r_1', b_2), (b_2, r_2, b_3), ..., (b_p, r_q', b)$. Indeed, the only triples that need to be considered in addition to the triples $(a, r_1, a_2), (a_2, r_2, a_3), ..., (a_p, r_p, b)$ are of the form $(a_i, eq, a_i)$, which we have assumed to belong to $\mathcal{G}$ for every $a_i \in \mathcal{E}$. For every path of type $r_1'; ...; r_q'$ whose *eq*-reduced type is $r_1; ...; r_p$, we thus find entirely similarly to before that

$$\mu_{r_1'; ...; r_q'}(\mathbf{a^{(i)}}) \preccurlyeq \mathbf{b^{(i+p)}}$$

Because of Lemma 3, this implies

$$\mu_r(\mathbf{a^{(i)}}) \preccurlyeq \mathbf{b^{(i+p)}}$$

In particular, we have

$$\mu_r(\mathbf{a^{(m)}}) \preccurlyeq \mathbf{b^{(m+p)}}$$

and because of the assumption that the GNN has converged after $m$ steps, we also have $\mu_r(\mathbf{a^{(m)}}) \preccurlyeq \mathbf{b^{(m)}}$. □

For $e \in \mathcal{E}$, let $paths_{\mathcal{G}}(e)$ be the set of all paths in the knowledge graph $\mathcal{G}$ which end in $e$. For a path $\pi$ in $paths_{\mathcal{G}}(e)$, we write $head(\pi)$ for the entity where the path starts and $rels(\pi)$ for the corresponding sequence of relations. For an entity $e$, we write $emb_m(e)$ for its embedding in layer $m$, i.e. $emb_m(e) = \mathbf{e^{(m)}}$. The following observation follows immediately from the construction of the GNN, together with Lemma 2.

**Lemma 4.** *For any entity $e \in \mathcal{E}$ it holds that*

$$\mathbf{e^{(m)}} \preceq \max\left(\mathbf{e^{(0)}}, \max_{\pi \in paths_{\mathcal{G}}(e)} \mu_{rels(\pi)}\left(emb_0(head(\pi))\right)\right)$$

We will also need the following technical lemma.

**Lemma 5.** *Suppose $\mathcal{P} \cup \mathcal{G} \not\models (a, r, b)$. Then there is some $i \in \{1, ..., \ell\}$ such that:*

- *$Z_i \subseteq I_r$; and*

- *whenever $\pi \in paths_{\mathcal{G}}(b)$ with $head(\pi) = a$, it holds that $I_{rels(\pi)} \cap Z_i = \emptyset$.*

*Proof.* Let us write $\mathcal{Z}_r = \{i \in \{1, ..., \ell\} \mid Z_i \subseteq I_r^1\}$. Note that $i \in \mathcal{Z}_r$ iff node $n_i$ in $\mathcal{H}$ has an incoming $r$-edge. It thus follows from condition (R1) that $\mathcal{Z}_r \neq \emptyset$. Suppose that for every $i \in \mathcal{Z}_r$, there was some $\pi \in paths_{\mathcal{G}}(b)$ with $head(\pi) = a$ such that $I_{rels(\pi)} \cap Z_i \neq \emptyset$. Let us write $X = \{rels(\pi) \mid \pi \in paths_{\mathcal{G}}(b), head(\pi) = a, I_{rels(\pi)} \cap Z_i \neq \emptyset\}$. We then have that for every $r$-edge in $\mathcal{H}$, there is a path $\tau$ connecting the same nodes, with $rels(\tau) \in X$. From Condition (R4), it then follows that $\mathcal{P} \cup \mathcal{G} \models (a, r, b)$, a contradiction. □

The following result shows that the GNN is unlikely to predict triples that cannot be inferred from $\mathcal{G} \cup \mathcal{P}$, as long as the embeddings are sufficiently high-dimensional.

**Proposition 9.** *Let $\mathcal{P}$ be a rule base and $\mathcal{G}$ a knowledge graph. Let $\mathcal{H}$ be a rule graph for $\mathcal{P}$ and let $\mathbf{Z}_{\mathbf{e}}^{(1)}$ be the entity representations that are learned by the corresponding GNN. For any $\varepsilon > 0$, there exists some $k_0 \in \mathbb{N}$ such that, when $k \geq k_0$, for any $m \in \mathbb{N}$ and $(a, r, b) \in \mathcal{E} \times \mathcal{R} \times \mathcal{E}$ such that $\mathcal{P} \cup \mathcal{G} \not\models (a, r, b)$, we have*

$$Pr[\mathbf{B_r Z_a^{(m)}} \preceq \mathbf{Z_b^{(m)}}] \leq \varepsilon$$

*Proof.* First, note that because of Lemma 1, what we need to show is equivalent to:

$$Pr[\mu_r(\mathbf{a^{(m)}}) \preceq \mathbf{b^{(m)}}] \leq \varepsilon$$

Let $(a, b) \in \mathcal{E} \times \mathcal{E}$ be such that $\mathcal{P} \cup \mathcal{G} \not\models (a, r, b)$. From Lemma 5, we know that there is some $i \in \{1, ..., \ell\}$ such that $Z_i \subseteq I_r^1$ and whenever $\pi \in paths_{\mathcal{G}}(b)$ with $head(\pi) = a$, it holds that $I_{rels(\pi)} \cap Z_i = \emptyset$. The following condition is clearly a necessary requirement for $\mu_r(\mathbf{a^{(m)}}) \preceq \mathbf{b^{(m)}}$:

$$\forall j \in Z_i \, . \, \mu_r(\mathbf{a^{(m)}}) \preccurlyeq_j \mathbf{b^{(m)}}$$

where we write $(x_1, ..., x_d) \preccurlyeq_j (y_1, ..., y_d)$ for $x_j \leq y_j$. We need in particular also that:

$$\forall j \in Z_i \, . \, \mu_r(\mathbf{a^{(0)}}) \preccurlyeq_j \mathbf{b^{(m)}}$$

Due to Lemma 4 this is equivalent to requiring that for every $j \in Z_i$ we have:

$$\mu_r(\mathbf{a^{(0)}}) \preccurlyeq_j \max\left(\mathbf{b^{(0)}}, \max_{\pi \in paths_{\mathcal{G}}(b)} \mu_{rels(\pi)}\left(emb_0(head(\pi))\right)\right)$$

We can view the coordinates of the input embeddings as random variables. The latter condition is thus equivalent to a condition of the following form:

$$\forall j \in Z_i \, . \, A_j^r \leq \max(B_j, X_j^1, ..., X_j^p)$$

where $A_j^r$ is the random variable corresponding to the $j^{\text{th}}$ coordinate of $\mu_r(\mathbf{a^{(0)}})$, $B_j$ is the $j^{\text{th}}$ coordinate of $\mathbf{b^{(0)}}$ and $X_j^1, ..., X_j^p$ are the random variables corresponding to the $j^{\text{th}}$ coordinate of the vectors $\mu_{rels(\pi)}\left(emb_0(head(\pi))\right)$. By construction, we have that the coordinates of different entity embeddings are sampled independently and that there are at least two distinct values that have a non-negative probability of being sampled for each coordinate. This means that there exists some value $\lambda > 0$ such that $Pr[A_j^r > B_j] \geq \lambda$ and $Pr[A_j^r > X_j^t] \geq \lambda$ for each $t \in \{1, ..., p\}$. Moreover, since we have that whenever $\pi \in paths_{\mathcal{G}}(b)$ with $head(\pi) = a$ it holds that $I_{rels(\pi)} \cap Z_i = \emptyset$, it follows that the random variable $A_j^r$ is not among $B_j, X_j^1, ..., X_j^p$. We thus have:

$$Pr[\forall j \in Z_i \, . \, A_j^r \leq \max(B_j, X_j^1, ..., X_j^p)]$$
$$\leq \left(1 - \lambda^{p+1}\right)^{|Z_i|}$$
$$= \left(1 - \lambda^{p+1}\right)^k$$
$$\leq e^{-k\lambda^{p+1}}$$

The value of $p$ is upper bounded by $\ell \cdot |\mathcal{E}|$, with $\ell$ the number of nodes in the rule graph. By choosing $k$ sufficiently large, we can thus make this probability arbitrarily small. In particular:

$$e^{-k\lambda^{p+1}} \leq \varepsilon \quad \Leftrightarrow \quad k \geq \frac{1}{\lambda^{p+1}} \log \frac{1}{\varepsilon}$$

$\square$

## B   CONSTRUCTING RULE GRAPHS

### B.1   PROOF OF PROPOSITION 3

Let $\mathcal{P}$ be a rule base which satisfies the conditions of Proposition 3, and let $r_1, ..., r_{|\mathcal{R}|}$ be the corresponding ranking of the relations. We construct a rule graph $\mathcal{H}$ for $\mathcal{P}$ as follows.

1. We add the node $n_0$.

2. For each relation $r \in \mathcal{R}$, we add a node $n_r$, and we connect $n_0$ to $n_r$ with an $r$-edge.

3. For $i$ going from $|\mathcal{R}|$ to 1:

   (a) For each rule $r_{j_1}(X_1, X_2) \wedge ... \wedge r_{j_q}(X_q, X_{q+1}) \to r_i(X_1, X_{q+1})$ with $r_i$ in the head and each $r_i$ edge between nodes $n$ and $n'$ in $\mathcal{H}$, we create fresh nodes $n_1, ..., n_q$ and add an $r_{j_1}$-link from $n$ to $n_1$, an $r_{j_2}$ link from $n_1$ to $n_2$, ..., an $r_{j_q}$-link from $n_q$ to $n'$.

Clearly the process terminates after a finite number of steps, noting that the new edges that are added for a rule $r_{j_1}(X_1, X_2) \wedge ... \wedge r_{j_q}(X_q, X_{q+1}) \to r_i(X_1, X_{q+1})$ cannot be $r_i$-edges, due to the assumption that $\mathcal{P}$ is free from cyclic dependencies. We also trivially have that condition (R1) is satisfied.

To see why (R2) is satisfied, first note that this is clearly the case after the first two steps have been completed. In the third step, when processing a rule $r_{j_1}(X_1, X_2) \wedge ... \wedge r_{j_q}(X_q, X_{q+1}) \to r_i(X_1, X_{q+1})$ and an edge from $n$ to $n'$, the only existing node where an incoming edge is added is $n'$ (where the other edges end in a fresh node). However, by construction, $n'$ can only have incoming $r_j$-edges with $j \geq i$ whereas $j_q < i$ because of the assumption that $\mathcal{P}$ is free from cyclic dependencies. The addition of the $r_{j_q}$-link from $n_q$ to $n'$ can thus not cause (R2) to become unsatisfied. It follows that (R2) still holds after the third step of the construction algorithm is finished.

Finally, the fact that (R3) and (R4) are satisfied straightforwardly follows from the construction.

## B.2 PROOF OF PROPOSITION 4

We write $\mathcal{R}_1$ for the set of relations that appear in the head of some rule from the considered rule base, and $\mathcal{R}_2 = \mathcal{R} \setminus \mathcal{R}_1$ for the remaining relations.

Let $\alpha(r_i) = r_i$ if $r_i \in \mathcal{R}_2$ and $\alpha(r_i) = \overline{r}_i$ otherwise. We clearly have that $\alpha(r_1)...\alpha(r_k) \in L_r$ iff $\mathcal{P}$ entails the following rule:

$$r_1(X_1, X_2) \wedge ... \wedge r_k(X_k, X_{k+1}) \to r(X_1, X_{k+1})$$

Since we have assumed that $\mathcal{P}$ has a rule graph, thanks to conditions (R3) and (R4), we can check whether this rule is valid by checking whether for each edge labelled with $r$ there is a path connecting the same nodes whose $eq$-reduced type is $r_1; ...; r_k$. Let $(n_i, n_j)$ be a an edge labelled with $r$. Then, we can construct a finite state machine (FSM) from $\mathcal{H}$ by treating $n_i$ as the start node and $n_j$ as the unique final node and interpreting $eq$ edges as $\varepsilon$-transitions (i.e. corresponding to the empty string). Clearly, this FSM will accept the string $r_1...r_k$ if there is a path labelled with $r_1; ...; r_k$ connecting $n_i$ to $n_j$. For each edge labelled with $r$, we can construct such an FSM. Let $F_1, ..., F_m$ be the languages associated with these FSMs. By construction, $L_r$ is the intersection of $F_1, ..., F_m$. Since $F_1, ..., F_m$ are regular, it follows that $L_r$ is regular as well.

## B.3 LEFT REGULAR RULE BASES

Given a left-regular rule base $\mathcal{P}$, we construct the corresponding rule graph $\mathcal{H}$ as follows.

1. We add the node $n_0$.

2. For each relation $r \in \mathcal{R}$, we add a node $n_r$, and we connect $n_0$ to $n_r$ with an $r$-edge.

3. For each rule of the form (10), we add an $r_2$-edge from $n_{r_1}$ to $n_{r_3}$.

4. For each node $n$ with multiple incoming $r$-edges for some $r \in \mathcal{R}$, we do the following. Let $\sharp_r$ be the number of incoming $r$-edges for node $n$. Let $p = \max_{r \in \mathcal{R}} \sharp_r$. We create fresh nodes $n_1, ..., n_{p-1}$ and add $eq$-edges from $n_i$ to $n_{i-1}$ ($i \in \{1, ..., p-1\}$), where we define $n_0 = n$. Let $r \in \mathcal{R}$ be such that $\sharp_r > 1$. Let $n'_0, ..., n'_q$ be the nodes with an $r$-link to $n$; then we have $q \leq p - 1$. For each $i \in \{1, ..., q\}$ we replace the edge from $n'_i$ to $n$ by an edge from $n'_i$ to $n_i$.

We now illustrate the construction process with an example.

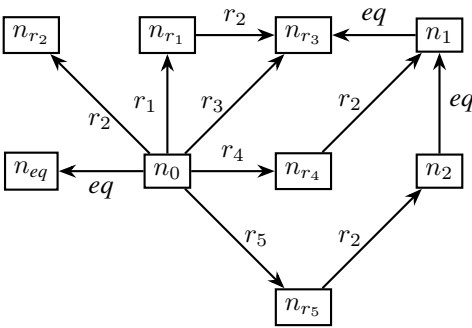

Figure 2: Rule graph for Example 5.

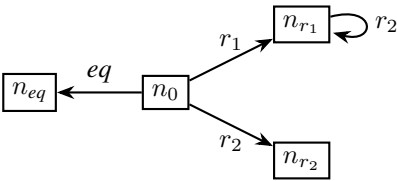

Figure 3: Rule graph for Example 6.

**Example 5.** *Let $\mathcal{P}$ contain the following rules:*

$$r_1(X, Y) \wedge r_2(Y, Z) \rightarrow r_3(X, Z)$$
$$r_4(X, Y) \wedge r_2(Y, Z) \rightarrow r_3(X, Z)$$
$$r_5(X, Y) \wedge r_2(Y, Z) \rightarrow r_3(X, Z)$$

*The corresponding rule graph is depicted in Figure 2. The nodes $n_1$ and $n_2$ were introduced in step 4 of the construction process. Before this step, there were $r_2$-edges from $n_{r_4}$ to $n_{r_3}$ and from $n_{r_5}$ to $n_{r_3}$. The node $n_{r_3}$ thus had three incoming $r_2$-edges, which violates condition (R2). This is addressed through the use of eq edges in step 4.*

Note that the rule graph may have loops, as illustrated next.

**Example 6.** *Let $\mathcal{P}$ contain the following rule:*

$$r_1(X, Y) \wedge r_2(Y, Z) \rightarrow r_1(X, Z)$$

*The corresponding rule graph is shown in Figure 3.*

The proposed construction process clearly terminates after a finite number of steps. To prove Proposition 5, we show that the proposed construction yields a valid rule graph for $\mathcal{P}$, i.e. that the resulting rule graph $\mathcal{H}$ satisfies (R1)–(R4).

The fact that (R1) is satisfied follows from the following lemma.

**Lemma 6.** *Let $\mathcal{P}$ be a left-regular set of closed path rules and let $\mathcal{H}$ be the graph obtained using the proposed construction method. For every $r \in \mathcal{R}$, it holds that $\mathcal{H}$ contains an outgoing $r$-edge from $n_0$.*

*Proof.* Let $r \in \mathcal{R}$. The edge from $n_0$ to $n_r$ is added in step 2 of the construction process. This edge may be removed in step 4, but in that case, a new $r$-edge is added from $n_0$ to a fresh node. □

The fact that (R2) is satisfied follows immediately from the construction in step 4. We now move to condition (R3).

**Lemma 7.** *Let $\mathcal{P}$ be a left-regular set of closed path rules and let $\mathcal{H}$ be the graph obtained using the proposed construction method. If $\mathcal{P}$ contains the rule $r_1(X_1, X_2) \wedge r_2(X_2, X_3) \rightarrow r_3(X_1, X_3)$, then whenever two nodes $n$ and $n'$ are connected in $\mathcal{H}$ by a path whose eq-reduced type is $r_3$, there is some node $n''$ such that $n$ and $n''$ are connected by a path whose eq-reduced type is $r_1$ and $n''$ and $n'$ are connected by a path whose eq-reduced type is $r_2$.*

*Proof.* The stated assertion clearly holds after step 3 of the construction method. Indeed, the only $r_3$-edge in $\mathcal{H}$ is from $n_0$ to $n_{r_3}$. Note in particular that no $r_3$ edges can be added in step 3, given our assumption that $\mathcal{P}$ is left-regular. Finally, it is also easy to see that this property remains satisfied after step 4. $\qquad\square$

The next lemma shows that (R3) is satisfied.

**Lemma 8.** *Let $\mathcal{P}$ be a left-regular set of closed path rules and let $\mathcal{H}$ be the graph obtained using the proposed construction method. Suppose nodes $n$ and $n'$ are connected with an edge of type $r$ and suppose $\mathcal{P} \models r_1(X_1, X_2) \wedge r_2(X_2, X_3) \wedge ... \wedge r_p(X_p, X_{p+1}) \to r(X_1, X_{p+1})$. Then there is a path whose eq-reduced type is $r_1; ...; r_p$ from $n$ to $n'$.*

*Proof.* Assume $\mathcal{P} \models r_1(X_1, X_2) \wedge r_2(X_2, X_3) \wedge ... \wedge r_p(X_p, X_{p+1}) \to r(X_1, X_{p+1})$. Let $n$ and $n'$ be nodes connected by an edge of type $r$. We show the result by structural induction. First, suppose $p = 2$. In this case, the considered rule is of the form $r_1(X_1, X_2) \wedge r_2(X_2, X_3) \to r(X_1, X_3)$. It then follows from Lemma 7 that there is a path whose *eq*-reduced type is $r_1; r_2$ connecting $n$ and $n'$. Let us now consider the inductive case. If $p > 3$ then $r_1(X_1, X_2) \wedge r_2(X_2, X_3) \wedge ... \wedge r_p(X_p, X_{p+1}) \to r(X_1, X_{p+1})$ is derived from at least two rules in $\mathcal{P}$ (given that the rules in $\mathcal{P}$ were restricted to have only two atoms in the body). The last step of the derivation of this rule is done by secting some rule $s_1(X, Y) \wedge s_2(Y, Z) \to r(X, Z)$ from $\mathcal{P}$ such that

$$\mathcal{P} \models r_1(X_1, X_2) \wedge ... \wedge r_{i-1}(X_{i-1}, X_i) \to s_1(X_1, X_i)$$
$$\mathcal{P} \models r_i(X_i, X_{i+1}) \wedge ... \wedge r_p(X_p, X_{p+1}) \to s_2(X_i, X_{p+1})$$

If there is a path from $n$ to $n'$ whose *eq*-reduced type is $r$, we know from Lemma 7 that there must be a path from $n$ to $n''$ with *eq*-reduced type $s_1$-edge and a path from $n''$ to $n'$ with *eq*-reduced type $s_2$, for some node $n''$ in $\mathcal{H}$. By induction, we furthermore know that there must then be a path with *eq*-reduced type $r_1; ...; r_{i-1}$ from $n$ to $n''$ and a path with *eq*-reduced type $r_i; ...; r_p$ from $n''$ to $n'$. Thus, we find that there must be a path with *eq*-reduced type $r_1; ...; r_p$ from $n$ to $n'$. $\qquad\square$

The fact that (R4) is satisfied follows from the next lemma.

**Lemma 9.** *Let $\mathcal{P}$ be a left-regular set of closed path rules and let $\mathcal{H}$ be the graph obtained using the proposed construction method. Suppose there is a path in $\mathcal{H}$ from $n_0$ to $n_r$ whose eq-reduced type is $r_1; ...; r_p$. Then it holds that $\mathcal{P} \models r_1(X_1, X_2) \wedge ... \wedge r_p(X_p, X_{p_1}) \to r(X_1, X_{p+1})$.*

*Proof.* The result clearly holds after step 2. We show that the result remains valid after each iteration of step 3. Suppose in step 3 we add an $r_2$-edge between $n_{r_1}$ and $n_{r_3}$. This means that:

$$\mathcal{P} \models r_1(X, Y) \wedge r_2(Y, Z) \to r_3(X, X)$$

Let $\tau$ be a path from $n_0$ to $n_r$. If $\tau$ does not contain the new $r_2$-edge, then the fact that the result is valid for $\tau$ follows by induction. Now, suppose that $\tau$ contains the new $r_2$ edge. Then $\tau$ is of the form $r_{i_1}; ...; r_{i_s}; r_2; r_{j_1}; ...; r_{j_t}$. By induction we have:

$$\mathcal{P} \models r_{i_1}(X_1, X_2) \wedge ... \wedge r_{i_s}(X_s, X_{s+1}) \to r_1(X_1, X_{s+1})$$

Clearly there is a path from $n_0$ to $n_{r_3}$ with *eq*-reduced type $r_3$. In particular, there is a path from $n_0$ to $n_{r_3}$ with *eq*-reduced type $r_3; r_{j_1}; ...r_{j_t}$. By induction, we thus have:

$$\mathcal{P} \models r_3(X_0, X_1) \wedge r_{j_1}(X_1, X_2) \wedge ...$$
$$\wedge r_{j_t}(X_t, X_{t_1}) \to r(X_0, X_{t+1})$$

Together we find that the stated result is satisfied.

Finally, we need to show that the result remains satisfied after step 4. This is clearly the case, as this step replaces edges of type $r$ with paths of type $r; eq; ...; eq$. The *eq*-reduced types of the paths from $n_0$ to $n_r$ thus remain unchanged after this step. $\qquad\square$

**Proposition 10.** *Let $\mathcal{P}$ be a left-regular set of closed path rules and let $\mathcal{H}$ be the graph obtained using the proposed construction method. It holds that $\mathcal{H}$ satisfies (R1)–(R4).*

*Proof.* The fact that (R1), (R3) and (R4) are satisfied follows immediately from Lemmas 6, 8 and 9. The fact that (R2) is satisfied follows trivially from the construction. $\qquad\square$

## B.4 BOUNDED INFERENCE

### B.4.1 PROOF OF PROPOSITION 6

Let $paths_{\mathcal{G}}^m(b)$ be the set of all paths in $\mathcal{G}$ of length at most $m$ which are ending in $b$.

**Lemma 10.** *For any entity $e \in \mathcal{E}$ it holds that*

$$\mathbf{e}^{(\mathbf{m})} \preceq \max\Big(\mathbf{e}^{(\mathbf{0})}, \max_{\pi \in paths_{\mathcal{G}}^m(e)} \mu_{rels(\pi)}\big(emb_0(head(\pi))\big)\Big)$$

*Proof.* This follows immediately from the construction of the GNN. $\square$

**Lemma 11.** *Let $\ell$ be the number of nodes in the given $m$-bounded rule graph. Suppose $\mathcal{P} \cup \mathcal{G} \not\models_m (a, r, b)$. Then there is some $i \in \{1, ..., \ell\}$ such that:*

- *$Z_i \subseteq I_r$; and*

- *whenever $\pi \in paths_{\mathcal{G}}^{m+1}(b)$ with $head(\pi) = a$, it holds that $I_{rels(\pi)} \cap Z_i = \emptyset$.*

*Proof.* This lemma is shown in exactly the same way as Lemma 5, simply replacing $paths_{\mathcal{G}}(b)$ by $paths_{\mathcal{G}}^{m+1}(b)$ and replacing Condition (R4) by Condition (R4m). $\square$

**Proposition 11.** *Let $\mathcal{P}$ be a rule base and $\mathcal{G}$ a knowledge graph. Let $\mathcal{H}$ be an $m$-bounded rule graph for $\mathcal{P}$ and let $\mathbf{Z}_{\mathbf{e}}^{(\mathbf{1})}$ be the entity representations that are learned by the corresponding GNN. For any $\varepsilon > 0$, there exists some $k_0 \in \mathbb{N}$ such that, when $k \geq k_0$, for any $i \leq m+1$ and $(a, r, b) \in \mathcal{E} \times \mathcal{R} \times \mathcal{E}$ such that $\mathcal{P} \cup \mathcal{G} \not\models_m (a, r, b)$, we have*

$$Pr[\mathbf{B_r Z_a^{(i)}} \preceq \mathbf{Z_b^{(i)}}] \leq \varepsilon$$

*Proof.* This result is shown in the same way as Proposition 2, by relying on Lemma 11 instead of Lemma 5. $\square$

### B.4.2 PROOF OF PROPOSITION 7

Given a set of closed path rules $\mathcal{P}$ we can construct an $m$-bounded rule graph as follows.

1. We add the node $n_0$.

2. For each relation $r \in \mathcal{R}$, we add a node $n_r$, and we connect $n_0$ to $n_r$ with an $r$-edge.

3. We repeat the following until convergence. Let $r \in \mathcal{R}$ and assume there is an $r$-edge from $n$ to $n'$. Let $r_1(X, Y) \wedge r_2(Y, Z) \to r(X, Z)$ be a rule from $\mathcal{P}$ and suppose that there is no $r_1; r_2$ path connecting $n$ and $n'$. Suppose furthermore that the edge $(n, n')$ is on some path from $n_0$ to a node $n_{r'}$, with $r' \in \mathcal{R}$ whose length is at most $m$. We add a fresh node $n''$ to the rule graph, an $r_1$-edge from $n$ to $n''$, and an $r_2$-edge from $n''$ to $n'$.

4. For each $r \in \mathcal{R}$ and $r$-edge $(n, n')$ such that for some rule $r_1(X, Y) \wedge r_2(Y, Z) \to r(X, Z)$ from $\mathcal{P}$ there is no $r_1; r_2$ path connecting $n$ and $n'$, we do the following:

   (a) We add a fresh node $n''$, an $r_1$-edge from $n$ to $n''$ and an $r_2$-edge from $n''$ to $n'$.

   (b) We repeat the following until convergence. For each $r'$-edge from $n$ to $n''$ and each rule $r_1'(X, Y) \wedge r_2'(Y, Z) \to r'(X, Z)$ from $\mathcal{P}$, we add an $r_1'$ edge from $n$ to $n''$ and an $r_2'$-loop to $n''$ (if no such edges/loops exist yet).

   (c) We repeat the following until convergence. For each $r'$-edge from $n''$ to $n'$ and each rule $r_1'(X, Y) \wedge r_2'(Y, Z) \to r'(X, Z)$ from $\mathcal{P}$, we add an $r_1'$-loop to $n''$ and an $r_2'$-edge from $n''$ to $n'$ (if no such edges/loops exist yet).

   (d) We repeat the following until convergence. For each $r'$-loop at $n''$, and each rule $r_1'(X, Y) \wedge r_2'(Y, Z) \to r'(X, Z)$ from $\mathcal{P}$, we add an $r_1'$-loop and an $r_2'$-loop to $n''$ (if no such loops exist yet).

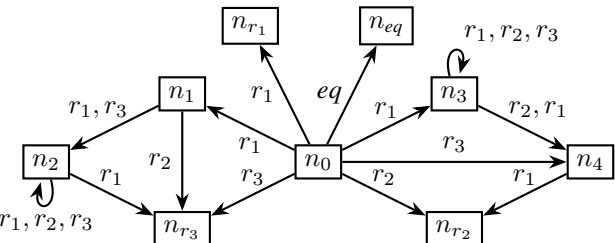

Figure 4: Rule graph for Example 7.

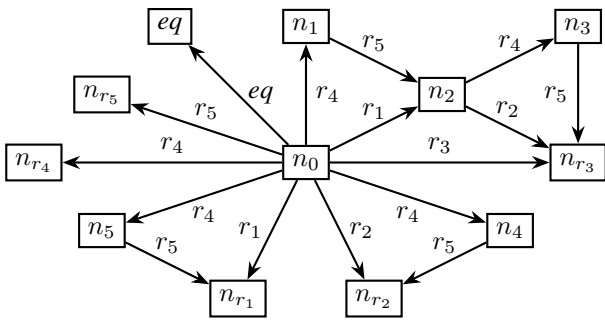

Figure 5: Rule graph for Example 8.

5. For each node $n$ with multiple incoming $r$-edges for one or more relations from $\mathcal{R}$, we do the following. Let $\sharp_r$ be the number of incoming $r$-edges for node $n$. Let $p = \max_{r \in \mathcal{R}} \sharp_r$. We create fresh nodes $n_1, ..., n_{p-1}$ and add $eq$-edges from $n_i$ to $n_{i-1}$ ($i \in \{1, ..., p-1\}$), where we define $n_0 = n$. Let $r \in \mathcal{R}$ be such that $\sharp_r > 1$. Let $n'_0, ..., n'_q$ be the nodes with an $r$-link to $n$; then we have $q \le p - 1$. For each $i \in \{1, ..., q\}$ we replace the edge from $n'_i$ to $n$ by an edge from $n'_i$ to $n_i$.

We illustrate the construction process with two examples.

**Example 7.** *Let us consider the following set of rules:*

$$r_1(X, Y) \wedge r_2(Y, Z) \to r_3(X, Z)$$
$$r_3(X, Y) \wedge r_1(Y, Z) \to r_2(X, Z)$$

*The corresponding $1$-bounded rule graph is shown in Fig. 4.*

**Example 8.** *Let us consider the following set of rules:*

$$r_1(X, Y) \wedge r_2(Y, Z) \to r_3(X, Z)$$
$$r_4(X, Y) \wedge r_5(Y, Z) \to r_1(X, Z)$$
$$r_4(X, Y) \wedge r_5(Y, Z) \to r_2(X, Z)$$

*The corresponding $2$-bounded rule graph is shown in Fig. 5. Note how this graph is in fact also a rule graph: due to the fact that there are no cyclic dependencies in the rule base $\mathcal{P} \cup \mathcal{G} \models_2 (e, r, g)$ is equivalent with $\mathcal{P} \cup \mathcal{G} \models (e, r, g)$.*

The construction process clearly terminates after a finite number of steps. Indeed, only edges that are on a path of length $m$ are expanded in step 3, and given that there are only finitely many such paths, step 3 must terminate. It is also straightforward to see that the other steps must terminate. We now show that the construction process yields a valid $m$-bounded rule graph.

Conditions (R1) and (R2) are clearly satisfied. Next, we show that condition (R3) is satisfied.

**Lemma 12.** *Let $\mathcal{P}$ be a set of closed path rules and let $\mathcal{H}$ be the resulting $m$-bounded rule graph, constructed using the proposed process. Suppose nodes $n$ and $n'$ are connected with an edge of type $r$ and suppose $\mathcal{P} \models r_{i_1}(X_1, X_2) \wedge r_{i_2}(X_2, X_3) \wedge ... \wedge r_{i_p}(X_p, X_{p+1}) \to r(X_1, X_{p+1})$. Then there is a path connecting $n$ to $n'$, whose eq-reduced type is $r_{i_1}; ...; r_{i_p}$.*

*Proof.* First, we show that at the end of step 4, there must be a path of type $r_{i_1}; ...; r_{i_p}$ connecting $n$ and $n'$. By construction, we immediately have that whenever two nodes $(n, n')$ are connected with an $r_i$-edge and $\mathcal{P}$ contains the rule $r_j(X, Y) \wedge r_l(Y, Z) \to r_i(X, Z)$ it holds that there exists some node $n''$ such that there is an $r_j$-edge from $n$ to $n''$ and an $r_l$ edge from $n''$ to $n'$. The existence of a path of type $r_{i_1}; ...; r_{i_p}$ then follows in the same way as in the proof of Lemma 8. It remains to be shown that the proposition remains valid after step 5. However, the paths in the final graph are those that can be found in the graph after step 4, with the possible addition of some *eq*-edges. This means in particular that after step 5, there must still be a path from $n$ to $n'$ whose *eq*-reduced type is $r_{i_1}; ...; r_{i_p}$. □

Finally, the fact that (R4m) is satisfied follows from the following lemma.

**Lemma 13.** *Let $\mathcal{P}$ be a set of closed path rules, and let $\mathcal{H}$ be the resulting $m$-bounded rule graph, constructed using the process outlined above. Suppose there is a path from $n_0$ to $n_r$ whose eq-reduced type if $r_1; ...; r_p$, with $p \leq m+1$. Then it holds that $\mathcal{P} \models r_1(X_1, X_2) \wedge ... \wedge r_p(X_p, X_{p_1}) \to r(X_1, X_{p+1})$.*

*Proof.* We clearly have that the proposition holds after step 3 of the construction method. After step 3, if there is an $r$-link between nodes $n$ and $n'$ and a rule $r_1(X, Y) \wedge r_2(Y, Z) \to r(X, Z)$ such that $n$ and $n'$ are not connected by an $r_1; r_2$ path, it must be the case that any path from $n_0$ to some node $n_r$ which contains the edge $(n, n')$ must have a length of at least $m + 1$. It follows that any path from $n_0$ to some node $n_r$ which contains an edge that was added during step 4 must have length at least $m + 2$. We thus have in particular that the proposition still holds after step 4. The paths in the final graph are those that can be found in the graph after step 4, with the possible addition of some *eq*-edges. Since the proposition only depends on the *eq*-reduced types of the paths, the result still holds after step 5. □

Together, we have shown the following result.

**Proposition 12.** *Let $\mathcal{P}$ be a set of closed path rules and let $\mathcal{H}$ be the graph obtained using the proposed construction method for $m$-bounded rule graphs. It holds that $\mathcal{H}$ satisfies (R1)–(R3) and (R4m).*

## C    EXPERIMENTAL DETAILS

This section lists additional details about our experiment's setup, benchmark datasets, and evaluation protocol. Section C.1 discusses further details of RESHUFFLE, while Section C.2 some additional implementation details. The origins and licenses of the standard benchmarks for inductive KGC are discussed in Section C.3. Details on RESHUFFLE's hyper-parameter optimisation are discussed in Section C.4. Finally, details about the evaluation protocol, together with the complete evaluation results, are provided in Section C.5.

### C.1    MODEL DETAILS

To initialise the entity embeddings, we set each coordinate to 0 or 1, with 50% probability. To train the model, we use the following scoring function for a given triple $(e, r, f)$:

$$s(e, r, f) = -\|\text{ReLU}(\mathbf{B_r Z_e^{(m)}} - \mathbf{Z_f^{(m)}})\|_2$$

where $m$ denotes the number of GNN layers. Note that $s(e, r, f) = 0$ reaches its maximal value of 0 iff $\mathbf{B_r Z_e^{(m)}} \preceq \mathbf{Z_f^{(m)}}$. For each $(e, r, f) \in \mathcal{G}$ we add an inverse triple $(f, r_{inv}, e)$ to $\mathcal{G}$. For each entity $e$, we also add the triple $(e, eq, e)$ to $\mathcal{G}$. Following the literature (Teru et al., 2020; Zhu et al., 2021), RESHUFFLE's training process uses negative sampling under the partial completeness assumption (PCA) (Galárraga et al., 2013), i.e., for each training triple $(e, r, f) \in \mathcal{G}$, $N$ triples (negative samples) are created by replacing $e$ or $f$ in $(e, r, f)$ by randomly sampled entities $e', f' \in \mathcal{E}$. To train RESHUFFLE, we minimise the margin ranking loss, defined as follows:

$$L(e, r, f) = \sum_{i=1}^{N} \max(0, s(e_i', r, f_i') - s(e, r, f) + \lambda) \tag{11}$$

Table 3: Number of relation, entities, and triples of the train, validation, and test split of the training and testing graph of the inductive benchmarks, split by corresponding benchmark versions v1-4.

|  |  | $\mathcal{R}_{Train}$ | $\mathcal{E}_{Train}$ | $\mathcal{G}_{Train}$ | $\mathcal{R}_{Test}$ | $\mathcal{E}_{Test}$ | $\mathcal{G}_{Test}$ |
|---|---|---|---|---|---|---|---|
| FB15k-237 | v1 | 180 | 1594 | 5226 | 142 | 1093 | 2404 |
|  | v2 | 200 | 2608 | 12085 | 172 | 1660 | 5092 |
|  | v3 | 215 | 3668 | 22394 | 183 | 2501 | 9137 |
|  | v4 | 219 | 4707 | 33916 | 200 | 3051 | 14554 |
| WN18RR | v1 | 9 | 2746 | 6678 | 8 | 922 | 1991 |
|  | v2 | 10 | 6954 | 18968 | 10 | 2757 | 4863 |
|  | v3 | 11 | 12078 | 32150 | 11 | 5084 | 7470 |
|  | v4 | 9 | 3861 | 9842 | 9 | 7084 | 15157 |
| NELL-995 | v1 | 14 | 3103 | 5540 | 14 | 225 | 1034 |
|  | v2 | 88 | 2564 | 10109 | 79 | 2086 | 5521 |
|  | v3 | 142 | 4647 | 20117 | 122 | 3566 | 9668 |
|  | v4 | 76 | 2092 | 9289 | 61 | 2795 | 8520 |

where $(e_i', r, f_i')$ is the i$^{\text{th}}$ negative sample and $\lambda > 0$ is a hyper-parameter, called the margin. At an intuitive level, the margin ranking loss pushes scores of true triples (i.e., those within the training graph) to be larger by at least $\lambda$ than the scores of triples that are likely false (i.e., negative samples).

## C.2 IMPLEMENTATION DETAILS

RESHUFFLE is trained on an NVIDIA Tesla V100 PCIe 32 GB GPU. We train RESHUFFLE for up to 1000 epochs, minimizing the margin ranking loss (see Equation 11) with the Adam optimiser (Kingma & Ba, 2015). If the Hits@10 score on the validation split of $\mathcal{G}_{Train}$ does not increase by at least $1\%$ within 100 epochs, we stop the training early.

RESHUFFLE was implemented using the Python library PyKEEN 1.10.1 (Ali et al., 2021). PyKEEN employs the MIT license and offers numerous benchmarks for KGC, facilitating the comfortable reuse of RESHUFFLE's code for upcoming applications and comparisons. Upon acceptance of our paper, we will provide RESHUFFLE's source code in a public GitHub repository to further facilitate the reuse of RESHUFFLE by our community.

## C.3 BENCHMARKS

Table 3 states the entity, relation, and triple counts of the training and test graphs, for each of the considered benchmarks.

We did not find a license for any of the three inductive benchmarks nor their corresponding transductive supersets. Furthermore, WN18RR is a subset of the WordNet database (Miller, 1995), which states lexical relations of English words. We also did not find a license for this dataset. FB15k-237 is a subset of FB15k (Bordes et al., 2013), which is a subset of Freebase (Toutanova & Chen, 2015), a collaborative database that contains general knowledge, such as about celebrities and awards, in English. We did not find a license for FB15k-237 but found that FB15k (Bordes et al., 2013) uses the CC BY 2.5 license. Finally, NELL-995 (Xiong et al., 2017) is a subset of NELL (Carlson et al., 2010), a dataset that was extracted from semi-structured and natural-language data on the web and that includes information about e.g., cities, companies, and sports teams. Also for NELL, we did not find any license information.

## C.4 HYPER-PARAMETER OPTIMISATION

Following Teru et al. (2020), we manually tune RESHUFFLE's hyper-parameters on the validation split of $\mathcal{G}_{Train}$. We use the following ranges for the hyperparameters: the number of RESHUFFLE's layers #Layers $\in \{3, 4, 5\}$, the embedding dimensionality parameters $l \in \{20, 25, 30\}$ and $k \in \{40, 60, 80\}$, the loss margin $\lambda \in \{0.5, 1.0, 2.0\}$, and finally the learning rate $lr \in \{0.005, 0.01\}$. We use the same batch and negative sampling size for all runs. In particular, we set the batch size to 1024 and the negative sampling size to 100. We report the best hyper-parameters for RESHUFFLE

Table 4: RESHUFFLE's best-performing hyper-parameters on FB15k-237 v1-4, WN18RR v1-4, and NELL-995 v1-4.

| | | #Layers | $l$ | $k$ | $\lambda$ | lr |
|---|---|---|---|---|---|---|
| FB15k-237 | v1 | 4 | 25 | 80 | 2.0 | 0.005 |
| | v2 | 3 | 30 | 60 | 1.0 | 0.005 |
| | v3 | 5 | 25 | 40 | 0.5 | 0.005 |
| | v4 | 3 | 30 | 80 | 1.0 | 0.01 |
| WN18RR | v1 | 3 | 20 | 40 | 1.0 | 0.01 |
| | v2 | 3 | 20 | 60 | 0.5 | 0.01 |
| | v3 | 3 | 20 | 40 | 1.0 | 0.01 |
| | v4 | 3 | 30 | 80 | 1.0 | 0.01 |
| NELL-995 | v1 | 3 | 20 | 80 | 2.0 | 0.005 |
| | v2 | 4 | 30 | 60 | 2.0 | 0.01 |
| | v3 | 4 | 25 | 40 | 0.5 | 0.01 |
| | v4 | 4 | 30 | 60 | 1.0 | 0.01 |

Table 5: RESHUFFLE's benchmark Hits@10 scores on all seeds together with the mean (*mean*) and standard deviation (*stdv*) of Hits@10.

| | FB15k-237 | | | | WN18RR | | | | NELL-995 | | | |
|---|---|---|---|---|---|---|---|---|---|---|---|---|
| | v1 | v2 | v3 | v4 | v1 | v2 | v3 | v4 | v1 | v2 | v3 | v4 |
| Seed 1 | 0.751 | 0.879 | 0.905 | 0.918 | 0.713 | 0.727 | 0.614 | 0.693 | 0.630 | 0.874 | 0.871 | 0.816 |
| Seed 2 | 0.744 | 0.892 | 0.908 | 0.916 | 0.707 | 0.726 | 0.574 | 0.690 | 0.650 | 0.860 | 0.893 | 0.808 |
| Seed 3 | 0.746 | 0.883 | 0.897 | 0.918 | 0.710 | 0.736 | 0.617 | 0.698 | 0.635 | 0.848 | 0.881 | 0.812 |
| *mean* | 0.747 | 0.885 | 0.903 | 0.918 | 0.710 | 0.729 | 0.602 | 0.694 | 0.638 | 0.861 | 0.882 | 0.812 |
| *stdv* | 0.004 | 0.007 | 0.005 | 0.001 | 0.003 | 0.006 | 0.024 | 0.004 | 0.010 | 0.013 | 0.011 | 0.004 |

split by each inductive benchmark in Table 4. Finally, we reuse the same hyper-parameters for each of RESHUFFLE's ablations, namely, RESHUFFLE$_{nL}$ and RESHUFFLE$^2$.

## C.5 EVALUATION PROTOCOL AND COMPLETE RESULTS

Following the standard evaluation protocol for inductive KGC, introduced by Teru et al. (2020), we evaluate RESHUFFLE's final performance on the test split of the testing graph by measuring the ranking quality of any test triple $r(e, f)$ over 50 randomly sampled entities $e'_i \in \mathcal{E}$ and $f'_i \in \mathcal{E}$: $r(e'_i, f)$ and $r(e, f'_i)$ for all $1 \leq i \leq 50$. Following Teru et al. (2020), we report the Hits@10 metric, i.e., the proportion of true triples (those within the test split of the testing graph) among the predicted triples whose rank is at most 10.

Table 5 states RESHUFFLE's benchmark results over all inductive datasets, as well as their means and standard deviations.

