# OpenReview forum: "Differentiable Reasoning about Knowledge Graphs with Reshuffled Embeddings"
_ICLR.cc/2025/Conference — Submitted to ICLR 2025_

### Official Review · Reviewer_7cYB · 2024-10-31

**Soundness:** 3
**Presentation:** 4
**Contribution:** 3
**Rating:** 6
**Confidence:** 3

**Summary:**

This paper is motivated by the shortcomings of region-based KG embedding methods, it is proposed to take advantage of the ordering constraints of the reshuffled entity embeddings. In practice, an implementation using GNNs is proposed to enable the ability to perform inductive KGC.

**Strengths:**

1. The structure of the paper is well organized, i.e., the scope of the problem is precisely stated, the motivation and related work are comprehensively discussed.
2. The formal presentation of the problem (e.g., examples), the theoretical investigation, and the proposal are satisfactory and very helpful for understanding the idea of the paper.
3. Experimental results on inductive KGC and ablation studies are promising.

**Weaknesses:**

1. $\mathcal{P}\cup \mathcal{G} \models (e,r,f)$ is unclear to me, i.e. the first paragraph on page 4. Could you please provide a concrete example illustrating how $it is applied? Also, please clarify whether the first 'and' in line 163 should be 'or' instead?
2. The order of entities is not formally defined before Equ. (3). It's better to be illustrated in section 3.
3. The computational cost of constructing rule graphs should be studied and compared with other rule-based KG embedding methods. For instance, the complexity of constructing the rule graph w.r.t. the number of entities or the time cost proportional to the training time. And the training time cost comparisons to RuleN, AnyBURL, Neural-LP, DRUM, including the possible rule construction procedure.
4. The experimental results of applying possible sparsity constraints should be interesting to add, as claimed in line 238, even if some technical obstacles hinder the applausability of the idea. And the ability to capture cyclic rules should be investigated in experiments, for example, some simulation experiments should be helpful, which is not shown in the existing real data concentrated evaluations.

**Questions:**

1. Regarding the definition of closed path rules, given a graph $Z'\rightarrow Y'\rightarrow X \rightarrow Y \rightarrow Z$, and $X \rightarrow Z$, is the closed path rule embodied? Since it is captured by the induced subgraph of $\{X,Y,Z\}$, but not by $\{X, Y', Z'\}$. Or the rules are fully represented at the instance level, i.e. the symbols used to denote a rule all correspond to entities and relations in the graph, instead of just being placeholders.
2. In Eq. (3), is the period in the conditional really a comma?
3. What are the justifications for the conditions when initiating the embeddings to learn from lines 209 to 212?

---

> ### Author Response · Authors · 2024-11-14
>
> We thank the reviewer for their detailed reading of our manuscript. Below we clarify some misunderstandings.
>
> **Weaknesses**
>
> W1
>
> We have $\mathcal{P}\cup \mathcal{G}\models (e,r,f)$ if the triple $(e,r,f)$ can be derived from the triples in the knowledge graph $\mathcal{G}$ using the rules in $\mathcal{P}$. This is formalised using a recursive definition:  $\mathcal{P}\cup \mathcal{G}\models (e,r,f)$ holds if either $(e,r,f)$ belongs to the knowledge graph $\mathcal{G}$, which is the base case, or there are triples $(e,r_1,e_2),...,(e_{p},r_p,f)$ which can be entailed and there is a rule in $\mathcal{P}$ which allows us to derive $(e,r,f)$ from $(e,r_1,e_2),...,(e_{p},r_p,f)$.
>
> For example, suppose $\mathcal{G} = \\{(e,r,f),(f,s,g),(g,t,h)\\}$ and $\mathcal{P}=\\{r(X,Y)\wedge s(Y,Z) \rightarrow u(X,Z), u(X,Y)\wedge t(Y,Z) \rightarrow v(X,Z)\\}$. Then we have  $\mathcal{P}\cup \mathcal{G}\models (e,r,f)$ and $\mathcal{P}\cup \mathcal{G}\models (f,s,g)$. Because of the rule $r(X,Y)\wedge s(Y,Z) \rightarrow u(X,Z)$ we then also have $\mathcal{P}\cup \mathcal{G}\models (e,u,g)$. Finally, together with the fact that $\mathcal{P}\cup \mathcal{G}\models (g,t,h)$, using the rule $u(X,Y)\wedge t(Y,Z) \rightarrow v(X,Z)$, we find $(e,v,h)$.
>
> The “and” on line 163 is correct (since we may need to use both triples from $\mathcal{G}$ and rule from $\mathcal{P}$ to derive the entailment).
>
> W2
>
> Section 3 focuses on the problem setting (independent of the specific choices of the considered model), whereas Section 4 introduces the model. It was not clear to us what aspects of Equation 3 the reviewer wants us to illustrate in Section 3.
>
> W3
>
> The rule graph is an abstraction of the proposed model, which we use for our theoretical analysis of the kind of rules that can be captured/learned. In practice, the set of rules $\mathcal{P}$ is not given and, accordingly, we never explicitly construct the rule graph. Note that, due to the close correspondence between the rule graph and the matrices $\mathbf{B_r}$ (as detailed in Equation 9), we can always trivially obtain the rule graph from the trained model. However, since in practice we learn the matrices $\mathbf{B_r}$ using a softmax, this process would give us a “soft approximation” of a rule graph.
>
> W4
>
> As mentioned on line 238, none of the techniques we considered for imposing sparsity led to improvements over the basic softmax formulation. As these methods did not seem sufficiently promising, we did not evaluate them over the full set of benchmarks. Among others, we experimented with techniques for adding a temperature to the softmax, including different schedules for adjusting the temperature after each epoch; formulations based on the Gumbel-softmax; and regularisation strategies.
>
> For the experiments, our main focus was to show that it is feasible to learn the parameters of our model, and to show that the model can be competitive with (differentiable) rule based methods. This is because we want to ensure that the models we are studying remain “practical”, even though our primary contribution is clearly on the theoretical understanding of the expressivity of KG embeddings. While it might be possible to find synthetic settings where the proposed model outperforms other variants, the main interest for future work would be to close the gap between the current performance and that of state-of-the-art methods. As we briefly discussed in the conclusions section, we believe that an important next step for future work will be to develop principled methods for non-monotonic or probabilistic reasoning with embeddings.

---

> ### Author Response · Authors · 2024-11-14
>
> **Questions**
>
> Q1
>
> Closed path rules denote dependencies between the relations that appear in the knowledge graph. Note that these rules are thus independent of the entities. For instance, a rule such as $r(X,Y)\wedge s(Y,Z) \rightarrow t(X,Z)$ means that whenever we have that $(e,r,f)$ and $(f,s,g)$ are valid triples, for any entities $e,r,f$, we also have that $(e,t,g)$ is a valid triple. The arguments of the relations in a rule represent variables (which we believe might be what the reviewer refers to as placeholders).
>
> Q2
>
> Different notations are commonly used to express quantified expressions. While commas are sometimes used to separate the quantified variables from the body of the expression, we prefer using periods as this causes less confusion in some cases (e.g. when enumerating conditions).
>
> Q3
>
> We need to ensure that there are at least two distinct values that have a non-zero probability of being sampled, since otherwise all entities would be represented in the same way. The assumption that different coordinates are sampled independently ensures that we can make the probability that $\mathbf{B_r^{(l)}}\mathbf{Z_e} \preccurlyeq \mathbf{Z_f^{(l)}}$ holds by chance arbitrarily small, by increasing the dimensionality of the embeddings if needed (see the proof of Proposition 2 for details).  We sample non-negative coordinates to make sure that the representation of a relation r, encoded in the matrix $\mathbf{B}_r$, is able to encode that a given coordinate is not important for that relation. We will clarify this in the paper.

---

> > ### Comment · Reviewer_7cYB · 2024-12-03
> >
> > Dear authors,
> >
> > Thank you for your effort in addressing my concerns - most of the confusions on the presentations and notations are well-dressed. Methodologically, I mostly agree with Reviewer r3W8 on the limitation of the scenarios that this work could handle. But I'm keeping my score.
> >
> > Thank you,

---

### Official Review · Reviewer_2114 · 2024-11-01

**Soundness:** 2
**Presentation:** 1
**Contribution:** 1
**Rating:** 3
**Confidence:** 5

**Summary:**

This paper proposes RESHUFFLE, a novel model for differentiable reasoning over knowledge graphs that utilizes ordering constraints for improved embedding efficiency and expressiveness. RESHUFFLE can capture a broader class of rules than prior models and is especially effective for inductive KG completion using graph neural networks.

**Strengths:**

1. Proposes a unique, scalable model based on ordering constraints, extending the scope of rule capture beyond previous region-based approaches.
2. Employs a GNN framework to achieve efficient, update-friendly KG embeddings, advantageous for inductive tasks.
3. Demonstrates strong performance on benchmark datasets, outperforming some of existing reasoning models in some datasets.
4. Includes a rigorous theoretical and empirical analysis that supports the model's efficiency and robustness.

**Weaknesses:**

1. **Inability to Handle Noisy Data**: The RESHUFFLE model struggles significantly with noisy datasets like WN18RR, a major weakness for real-world applications where knowledge graphs often contain errors or incomplete data. The lack of adaptability to noisy or uncertain data undermines the model’s generalizability and limits its utility across varied datasets.

2. **Expressivity Limitations with Cyclic Dependencies**: Despite proposing a novel approach, RESHUFFLE cannot represent certain cyclic dependencies in rule bases, which restricts its applicability to only a subset of knowledge graphs. This limitation contradicts the paper's goal of providing a broadly applicable, flexible model for knowledge graph completion.

3. **Insufficient Support for Probabilistic Reasoning**: The model’s deterministic approach to embedding rules falls short in settings requiring probabilistic or non-monotonic reasoning, which are increasingly important in real-world KGs. This oversight limits the paper's contribution, as modern KGs benefit from methods that accommodate uncertain or evolving data.

4. **Over-Reliance on Ordering Constraints**: The novel use of ordering constraints, while interesting, imposes rigid structural assumptions that may lead to overfitting and limit the model’s expressivity. The reliance on strict ordering could restrict its effectiveness in scenarios requiring nuanced or flexible rule handling, making it less adaptable to complex inference tasks.

5. **Presentation and Organization Issues**. The paper's presentation and organization need improvement. The storyline, particularly in the introduction, lacks clarity and focus. Clearer structuring and a refined presentation of the motivation, methodology, and contributions would help readers follow the key points more easily.

6. **Performance Trade-offs with State-of-the-Art**: ESHUFFLE underperforms compared to NBFNet on multiple benchmarks where high accuracy is essential. Although RESHUFFLE offers efficiency advantages, its relatively lower accuracy raises concerns about its competitiveness. Additionally, A* Net [1], an incremental model of NBFNet, demonstrates efficiency and scalability; thus, the current experimental comparison with baselines appears insufficient.

7. **Limited and Incomplete Baselines**: The baselines included in this paper focus only on three categories—GNN-based, classical rule-based, and differentiable rule-based methods—overlooking several prominent classes, including translational KG embedding (e.g., RotatE[2], TransE[3], TransH[4]), probabilistic reasoning (e.g., Markov Logic Network[5], Probabilistic Soft Logic[6]), neuro-symbolic approaches (e.g., pLogicNet[7], RNNLogic[8], RLogic[9], DiffLogic[10]), and region-based models. Additionally, several recent state-of-the-art methods (e.g., A* Net[1], DiffLogic[10]) are missing, while RESHUFFLE still underperforms against NBFNet, a baseline from 2021. A more comprehensive and up-to-date set of baselines would strengthen the empirical claims.

8. **Insufficient experiments**. The experimental setup includes only a baseline study and an ablation study, which is insufficient to fully support the claims made in the paper regarding RESHUFFLE’s generalization ability and rule-capturing capabilities. Moreover, the evaluation is limited to Hit@10, whereas additional metrics like Mean Reciprocal Rank (MRR) and Hit@1, which offer more insight into ranking quality, are missing.


- [1] Zhu, Zhaocheng, et al. "A* net: A scalable path-based reasoning approach for knowledge graphs." Advances in Neural Information Processing Systems 36 (2024).
- [2] Sun, Zhiqing, et al. "Rotate: Knowledge graph embedding by relational rotation in complex space." arXiv preprint arXiv:1902.10197 (2019).
- [3] Bordes, Antoine, et al. "Translating embeddings for modeling multi-relational data." Advances in neural information processing systems 26 (2013).
- [4] Wang, Zhen, et al. "Knowledge graph embedding by translating on hyperplanes." Proceedings of the AAAI conference on artificial intelligence. Vol. 28. No. 1. 2014.
- [5] Richardson, Matthew, and Pedro Domingos. "Markov logic networks." Machine learning 62 (2006): 107-136.
- [6] Bach, Stephen H., et al. "Hinge-loss markov random fields and probabilistic soft logic." Journal of Machine Learning Research 18.109 (2017): 1-67.
- [7] Qu, Meng, and Jian Tang. "Probabilistic logic neural networks for reasoning." Advances in neural information processing systems 32 (2019).
- [8] Qu, Meng, et al. "Rnnlogic: Learning logic rules for reasoning on knowledge graphs." arXiv preprint arXiv:2010.04029 (2020).
- [9] Cheng, Kewei, et al. "Rlogic: Recursive logical rule learning from knowledge graphs." Proceedings of the 28th ACM SIGKDD Conference on Knowledge Discovery and Data Mining. 2022.
- [10] Shengyuan, Chen, et al. "Differentiable neuro-symbolic reasoning on large-scale knowledge graphs." Advances in Neural Information Processing Systems 36 (2024).

**Questions:**

Please see above weaknesses for my concerns.

---

> ### Author Response · Authors · 2024-11-14
>
> We thank the reviewer for the careful reading of our manuscript. However, we disagree with several of the claims that were made, and we hope that the reviewer will be willing to reconsider their assessment in light of our response.
>
> The reviewer criticises ReshufflE because of its limitations when it comes to handling cyclic dependencies. In fact, there are no existing models that can provably capture cyclic dependencies. The only exception is the theoretical construction from (Gutierrez-Basulto & Schockaert, 2018), which cannot be implemented as it requires learning arbitrary polytopes with exponentially many vertices. In fact, our approach is the first practical approach that is provably capable of modelling cyclic dependencies in the bounded reasoning setting. Moreover, our model is also the first that is provably capable of capturing a particular class of rule bases with cyclic dependencies in the general setting. We thus strongly believe that the ability to model cyclic dependencies is a key strength of our model, rather than a weakness.
>
> The reviewer also argues that ordering constraints may lead to overfitting. In fact, the opposite is true. While ordering constraints indeed impose a rigid structure, this is precisely the reason why we use ordering constraints and is actually aimed at preventing overfitting. Indeed, one of the core claims that we make in the paper is that the use of ordering constraints gives us a practical method for allowing cross-coordinate comparisons. The result for ReshufflE$^2$ in the ablation analysis confirms this, showing that a more general model significantly underperforms.
>
> Weaknesses 1 and 3 are points that we acknowledge in the conclusions section. Rule-based methods are indeed limited in their ability to deal with noise, and in cases where probabilistic reasoning is necessary. Our main aim in this paper has been to develop a better theoretical understanding of how rules can be modelled in a principled way using KG embedding models. Studying probabilistic reasoning would indeed be an important next step, but we cannot really do this before understanding how rules can be modelled. We strongly believe that the proposed model provides the foundations for developing a more general understanding of how expressive KG embedding models (and GNNs) can be designed, and that this presents a valuable contribution.
>
> Weaknesses 6-8 refer to the empirical analysis. Given the focus of our paper on developing a theoretical understanding, the aim of our experimental analysis was to show that the parameters of the proposed model can be learned in practice. We believe the experiments indeed show this, and furthermore show that the model is competitive against existing differentiable rule-based methods. At the same time, the experiments also show that the model underperforms the state-of-the-art, which is something that we have been clear on in the paper. Including more baselines would thus not affect our conclusions. As a minor point, please note that several of the suggested baselines under point 7 can only be used in the transductive setting, and are thus not applicable in our experiments.

---

> > ### Comment · Reviewer_2114 · 2024-11-22
> > **Response to authors**
> >
> > Thank you for the clarification. I appreciate that your work focuses on developing a theoretical understanding of how KG embeddings can enable rule learning, particularly differentiable rule learning. I have acknowledged your contributions in the strengths section.
> >
> > However, some of my concerns remain unaddressed. Below, I outline these concerns and hope the authors can address them in subsequent responses:
> >
> > **1.Rule-based methods and probabilistic reasoning**
> >
> > I disagree with the claim that "Rule-based methods are indeed limited in their ability to deal with noise and in cases where probabilistic reasoning is necessary." Existing rule-based methods such as Markov Logic Networks (MLN) and Probabilistic Soft Logic (PSL) already handle noise effectively:
> > - MLN is a well-established, principled method with efficient inference engines, as demonstrated in recent works like [2].
> > - PSL supports differentiable learning of weights and truth score assignments, enabling it to adapt to noisy settings [1].
> > - Additionally, neuro-symbolic methods incorporating weighted rules, such as [3], already provide differentiable reasoning on KGs. Probabilistic reasoning is not a "next step" but a longstanding focus in the field of knowledge base reasoning.
> >
> > **2.Experimental claims**
> >
> > The argument that "the aim of our experimental analysis was to show that the parameters of the proposed model can be learned in practice" is insufficient. Demonstrating that parameters are learnable is the minimum expectation for a top-tier machine learning conference and does not justify the theoretical claims in the paper.
> >
> > What is needed are experiments that:
> >
> > - Directly answer the research questions posed by your work.
> > - Provide explicit insights that justify your claims, supported by reproducible results.
> >
> > Merely showing parameter learnability does not align with the level of experimental rigor expected at this level.
> >
> > **3.Baseline coverage**
> >
> > The baselines included in the paper are limited in scope. While your work competes against differentiable rule-based methods, many other methods exploit rules for reasoning without requiring differentiability. The claim that some suggested methods can only be applied in a transductive setting is valid, but several others (e.g., NCRL, RLogic, A* Net) also support inductive relational learning.
> >
> > Additionally, inductive reasoning methods can often be applied in transductive settings. The key distinction in transductive reasoning is the absence of unseen entities or relations during testing. These points should be clarified, and the baseline comparisons should include methods relevant to both transductive and inductive reasoning.
> >
> >
> > **4. Experimental depth and insights**
> >
> > The experimental setup does not adequately support the claims of "generalization ability" and "rule-capturing capability."
> > Specifically:
> >
> > - **Performance**: Inferior results do not substantiate these claims.
> > - **Ablation studies**: The current ablations only show that each component contributes to overall effectiveness but fail to provide deeper insights into rule quality or utilization.
> >
> > More comprehensive experiments are required, such as:
> >
> > - Evaluating rule quality quantitatively (e.g., precision, coverage).
> > - Analyzing the distribution of mined rules and their impact on reasoning.
> > - Real ablation studies that demonstrate how different rule types enhance performance or generalization.
> >
> > **References**
> > - [1] Bach, Stephen H., et al. "Hinge-loss Markov random fields and probabilistic soft logic." Journal of Machine Learning Research 18.109 (2017): 1-67.
> > - [2] Fang, Huang, et al. "MLN4KB: An efficient Markov logic network engine for large-scale knowledge bases and structured logic rules." Web Conf. 2023.
> > - [3] Shengyuan, Chen, et al. "Differentiable neuro-symbolic reasoning on large-scale knowledge graphs." Advances in Neural Information Processing Systems 36 (2024).
> >
> > I look forward to your responses addressing these concerns.

---

> > > ### Author Response · Authors · 2024-11-22
> > >
> > > Thanks for your detailed response.
> > >
> > >
> > > **Rule-based methods and probabilistic reasoning**
> > >
> > >
> > > We are well-aware of the field of statistical relational learning (SLR), and we certainly didn’t want to claim that probabilistic reasoning in general is an unsolved problem. Our point is that the expressivity of KG embedding models is currently still poorly understood. Even for hard rules, we don’t have a full understanding of the kinds of rule bases that can be faithfully captured by different kinds of embedding models. However, what is clear already is that many of the most popular embedding models are severely limited in this respect. As a simple example, the DiffLogic paper that is cited by the reviewer relies on RotatE. But this model cannot even model basic composition rules, since it cannot distinguish a rule like $r_1(X,Y)\wedge r_2(Y,Z)\rightarrow r_3(X,Z)$ from the variant where the order of the relations in the body is swapped, i.e.  $r_2(X,Y)\wedge r_1(Y,Z)\rightarrow r_3(X,Z)$. Our work is devoted to a better understanding how we can design simple embedding models that are provably capable of modelling certain classes of rule bases.
> > >
> > >
> > > As an important line for future work, we would like to study the same question for probabilistic reasoning. For instance, is it possible to design a KG embedding model that is provably capable of modelling MLNs (e.g. in the sense that the triples that are captured by the embedding are exactly those that can be obtained using MAP-inference from the MLN)? To the best of our knowledge, this question has not yet been studied in the literature. Note that this is fundamentally different from the contributions made by DiffLogic, and related approaches such as pLogicNet, which use SRL models to help train a KG embedding. The latter approaches improve how the KG embeddings are trained, but they do not address the limited expressivity of the underlying KG embeddings themselves.
> > >
> > >
> > > **Experimental claims**
> > >
> > >
> > > If we don’t care about learnability, we already know that arbitrary sets of closed path rules can be captured, which was shown by Gutierrez-Basulto and Schockaert (2018). However, their construction allows relations to be modelled by arbitrary polytopes, and thus cannot be implemented in practice. Our central hypothesis in this paper is that cross-coordinate comparisons are necessary for escaping the limitations of earlier (practical) region based embedding models such as ExpressivE. However, this needs to be done in a careful way. First, bilinear models such as RESCAL allow cross-coordinate comparisons, but they cannot even faithfully capture hierarchy rules, which is why we propose to use order embeddings instead. Second, without a careful design, models with cross-coordinate comparisons are prone to overfitting (which is indeed also an important limitation of RESCAL). With this in mind, we tried to find the simplest possible model that can capture arbitrary sets of closed path rules (for bounded inference).
> > >
> > >
> > > Now, the central research questions that our experimental analysis tries to answer are as follows. While the model we propose is capable of capturing sets of rule bases in theory, is it actually capable of learning such rule bases in practice? We tried to design the model to be as simple as possible, but is it simple enough to avoid overfitting? In particular, since we cannot easily learn the sparse binary matrices that are used in the theoretical construction, we have to rely on a soft approximation instead. Can the resulting model still avoid overfitting? We believe that our analysis answers these questions, by showing that we can match (and in most cases outperform) rule-based methods on the task of inductive KG completion.
> > >
> > >
> > > **Baselines**
> > >
> > >
> > > Our analysis shows that our method’s performance is comparable to that of rule-based methods (as well as some differentiable rule based methods). We don’t claim that we can outperform state-of-the-art approaches, and we highlight several reasons why that is expected to be the case. While we can certainly add some additional baseline results to provide more context, we don’t believe that this would materially change the analysis.

---

> > > ### Author Response · Authors · 2024-11-22
> > >
> > > **Experimental depth and insights**
> > >
> > > The reviewer criticises the performance of the method. The ICLR reviewer guidelines are clear about the fact that not outperforming the state-of-the-art is not in itself a reason for rejection, stating that “Submissions bring value to the ICLR community when they convincingly demonstrate new, relevant, impactful knowledge. Submissions can achieve this without achieving state-of-the-art results.” (https://iclr.cc/Conferences/2025/ReviewerGuide, see FAQ at the bottom). Studying the expressivity of KG embedding models is a fundamental but highly challenging problem. We want to take a long-term view and try to understand, at a fundamental level, what is needed in an embedding model to allow it to capture certain types of knowledge. We believe that developing this research agenda is highly valuable, but meaningful progress would be seriously hampered if every paper is expected to demonstrate improved state-of-the-art results on top of developing theoretical results and insights.
> > >
> > > The suggestions about the need for additional experiments are about understanding the limitations of rule learning in general (e.g. the impact of mined rules on reasoning) and about better understanding the characteristics of the considered benchmarks (e.g. which kinds of rules are most important for performing well on these benchmarks). These questions are largely orthogonal to our focus in this paper. Moreover, while empirical papers may be expected to include analyses of this kind, we believe that our empirical analysis is sufficiently comprehensive given the theoretical nature of our contribution.

---

> > > > ### Comment · Reviewer_2114 · 2024-11-22
> > > > **Response to authors**
> > > >
> > > > Thank you for the prompt and detailed response.
> > > >
> > > > **I understand and have acknowledged the theoretical contributions of this work, as I stated many times**. However, I don’t think the concerns have been fully addressed through your responses. Furthermore, your responses have raised additional concerns. For instance:
> > > >
> > > > - **[New concern 1]**. In your response to the concern about baseline coverage, the claim "Our analysis shows that our method’s performance is comparable to that of rule-based methods" appears unreliable, as strong and expressive rule-based methods (e.g., A* Net, RLogic, NCRL) are not included in your baselines. This is exactly why we think the coverage of the baseline is insufficient.
> > > > - **[New concern 2]**. Concerns about the experimental results under transductive settings remain unaddressed, as you did not provide any clarification or analysis. Please do not ignore my concerns.
> > > >
> > > > Furthermore, You challenge me with the ICLR guidelines and argue the reasonability of the inferior performance of your work. I did not state that outperforming the SOTA is necessary. While other reviewers(UDqZ) worry about practicability, I understand that you implemented a minimum viable prototype model to demonstrate the practicability of your work. Instead, I supported you to address this concern by suggesting additional experimental results, where you may
> > > >
> > > > - Clearly demonstrate where the inferior performance comes from and provide explanations for the trade-offs involved.
> > > > - Show what makes your model effective in specific scenarios and explain why the observed performance is acceptable given the theoretical contributions.
> > > >
> > > > I believe additional experiments are also required by reviewer r3W8, who expressed "the authors can make their work more convincing by showing use cases where their theoretical model aligns well with empirical settings".
> > > >
> > > > Without these insights from experimental results, it becomes difficult to assess the empirical impact of your approach and the reliability of your theoretical conclusions in practical scenarios.
> > > >
> > > > Weighing the resolved, unresolved, and newly raised concerns, I have adjusted my rating accordingly.

---

> > > > > ### Author Response · Authors · 2024-11-22
> > > > >
> > > > > Thanks for the continued discussion.
> > > > >
> > > > > Regarding new concern 1: A\*Net is a variant of NBFNet, which is more efficient but underperforms the latter. We can add A\*Net results to the paper, but this would not change the analysis. RLogic and NCRL have, to our knowledge, not yet been evaluated on the inductive setting. Furthermore, the question of whether we can outperform all rule-based methods is not our primary concern. We accept that the empirical performance of our model is not at the level of the state-of-the-art, but we believe our experiments are sufficient considering the scope of the paper.
> > > > >
> > > > > Regarding new concern 2: We agree that an evaluation in the transductive setting would be possible in principle. We focus on the inductive setting in the paper, as this setting is better aligned with rule-based reasoning.
> > > > >
> > > > > Regarding the closing statement ("Without these insights from experimental results, it becomes difficult to assess the empirical impact of your approach and the reliability of your theoretical conclusions in practical scenarios"): We agree that more experiments would be needed if we wanted to demonstrate the empirical impact of our approach, but this is not the focus of our experiments. We accept that the model in its current form does not have a clear practical advantage over existing models. But we argue that our results are nonetheless significant as they further our understanding of what is possible, in terms of theoretical expressivity, with embeddings.

---

### Official Review · Reviewer_r3W8 · 2024-11-03

**Soundness:** 4
**Presentation:** 4
**Contribution:** 2
**Rating:** 5
**Confidence:** 4

**Summary:**

This paper proposes ReshufflE, a knowledge graph embedding model based on GNNs which aims to better capture closed path rules. To this end, ReshufflE proposes an order-based relation scoring function, in which a triple $(e, r, f)$ holds when $r$, which defines a mapping between head entity ($e$) and tail entity ($f$) dimension indices, is true for all mapped dimensions. As orderings are non-differentiable, the paper proposes a soft approximation to produce the ReshufflE model. The paper additionally proposes dimensionality reduction based on Kronecker product to reduce parametrization, and subsequently overfitting. The paper provides an extensive theoretical analysis of the rules that ReshufflE can capture, and in doing so defines the notion of a *rule graph*, showing that any set of rules representable by a rule graph can be captured by the ReshufflE GNN. The paper also provides negative results when rule graphs cannot be built, further reporting sets of rule grammars that correspond to the aforementioned rule graphs and subsequently can be captured. As a final theoretical contribution, the paper weakens the general reasoning requirement and looks into bounded reasoning paths, showing that an $m-$layer ReshufflE GNN can capture arbitrary sets of $m-$bounded rules.

On the empirical front, the paper runs experiments on standard inductive link prediction benchmarks, showing competitive performance with rule-based systems (with the exception of WN18RR), and reporting an ablation study on Reshuffle to validate the model's design choices.

**Strengths:**

- The theoretical arguments appear sound, and the problem being addressed is well-motivated: Representing closed path rules is an important question in knowledge graph embeddings, and this paper proposes an improvement over the state of the art.
- The discussion of the results is very objective, and limitations are acknowledged upfront. On the theoretical front, the set of closed path rules that can and cannot be captured is explored very thoroughly. Empirically, less flattering results are highlighted and discussed.
- The writing of this paper is easy to follow. I especially appreciate the examples and figures in the theoretical discussions, which really help clarify the contributions of this work.

**Weaknesses:**

- Unfortunately, the paper appears to be highly specialized to representing closed path rules, so much so that it sacrifices useful inductive biases that are often standard in other, more basic models. For example, ComplEx clearly fails to represent hierarchies in the general sense, but can learn that a relation $r$ is symmetric, i.e., $r(x, y) \implies $r(y, x). By contrast, reshuffle seems to specialize exclusively in closed path rules, at the expense of, e.g., symmetry. Concretely, if $r$ is a symmetric relation, then, in ReshufflE, $e_{\sigma_r} (i) \leq f$ and $f_{\sigma_r} (i) \leq e$ must hold simultaneously for all $i$ in $I_r$. However, this implies that symmetry cannot be enforced at the relation level, as the aforementioned inequalities are entity-dependent, preventing the model from capturing this pattern (this could be a reason for the underperformance on WN18RR, in my opinion, but I leave this as a comment for the authors to consider). Therefore, it would be very useful for the authors to discuss which of the basic inference patterns (symmetry, reflexiveness, hierarchy, etc.) can be captured by ReshufflE and how, in order to better place the strengths of ReshufflE more generally.

- The empirical performance of ReshufflE is not convincing, and seems to corroborate my above concern about over-specialization. To address this, I suggest that the authors try to report settings where ReshufflE can achieve SOTA results, to at least establish a set of rules of thumb where this model can be considered a strong SOTA candidate. Moreover, I recommend that the authors include more benchmarks, including potentially transductive benchmarks, to provide a more holistic picture of the strengths and weaknesses of this model.

Overall, I find that Reshuffle provides a meaningful and useful theoretical contribution that has a place in the knowledge graph embeddings literature. However, I have serious concerns about the over-specialization of this model, and how this seems to negatively affect representing common patterns such as symmetry, hierarchies, both separately and jointly with the aforementioned closed path rules. Note that over-specialization is not a problem in itself, as specialized models can be advantageous in the right settings: The problem here is that this hasn't been clearly demonstrated neither in a theoretical nor an empirical setting. Hence,  given my current concerns with the paper, I lean towards rejection. Nonetheless, I am happy to revise my rating should the authors address my concerns. In particular, my main recommendations to the authors to 1) better elaborate the strengths and weaknesses of the model with regards to other rules, and 2) to provide concrete empirical settings (including potential synthetic data, but ideally real-world data) in which Reshuffle has a compelling competitive advantage, to illustrate scenarios where the model can more reasonably be applied.

**Questions:**

Please see "Weaknesses" section above.

---

> ### Author Response · Authors · 2024-11-14
>
> We thank the reviewer for the detailed and constructive feedback.
>
> **Overspecialisation of the model**
>
> Modelling symmetry is actually possible with ReshufflE, thanks to the fact that we add inverse triples to the knowledge graph. For instance, if we have a triple $(e,r,f)$ in the original knowledge graph, then we will also include the triple $(f,r_{\text{inv}},e)$. To capture the fact that r is symmetric, the model then has to learn that the parameters of $r$ and $r_{\text{inv}}$ are the same.
>
> A hierarchy rule $r_1(X,Y)\rightarrow r_2(X,Y)$ can be captured by ensuring that every pair of nodes in the rule graph that is connected by an $r_1$-edge is also connected by an $r_2$-edge. Similarly, we can capture intersection rules such as $r_1(X,Y)\wedge r_2(X,Y)\rightarrow r(X,Y)$, by ensuring in the rule graph that every pair of nodes connected by an $r$-edge is also connected by an $r_1$-edge or by an $r_2$-edge. What is still an open question is whether arbitrary sets of intersection, hierarchy and closed path rules can be modelled jointly. Furthermore, in its current form, ReshufflE is also limited when it comes to modelling reflexivity. We will clarify these points in the paper.
>
> **Empirical performance**
>
> As for the empirical performance, we want to emphasize that our aim was to show that suitable model parameters can be learned in practice, and we would argue that our results are actually convincing in this respect, even if the performance is below the state-of-the-art. Our paper is the first work that introduces a practical model that can capture arbitrary sets of closed path rules (for bounded inference), as the only previous approach known to have this capability requires learning arbitrary polytopes (with exponentially many vertices).
>
> Our focus has clearly been on developing a theoretical understanding of how closed path rules  can be modelled. We believe that the current model provides the foundations for designing competitive KG completion models, but that achieving state-of-the-art results would require changes to the model. In particular, as the model outperforms existing rule-based approaches in most cases, we believe that future work should focus on better understanding the limitations of rule-based approaches in general. This could involve developing methods that can provably capture probabilistic logic programs, for instance (which would enable forms or reasoning that involve aggregating multiple pieces of weak evidence, as we highlighted in the conclusions). Doing this in a principled way is definitely a challenge, but we believe that developing this longer-term research agenda is important. However, this requires studying models that are not yet at the level of the state-of-the-art, which is the approach we took in this paper.

---

> ### Comment · Reviewer_r3W8 · 2024-11-15
> **Reviewer follow-up**
>
> I thank the authors for their response. I understand the motivation behind this work, which is why I suggested looking for more explicit settings highlighting its strength, as opposed to strictly finding state-of-the-art solutions. I agree with the authors that pursuing conceptually sound models is important. My point here is simply that the authors can make their work more convincing by showing use cases where their theoretical model aligns well with empirical settings, e.g., a synthetic dataset of closed path rules.
>
> On the rules front, I thank the authors for the clarification. However, I must clarify that inverse relations are a different (simpler) inference pattern to symmetry, and are separately studied in the literature. Defining an inverse relationship to capture symmetry does not mean symmetry is now captured. As for hierarchy, I believe that your point may hold for a finite number of rules, but will not generalize to arbitrary sets of hierarchy rules because of the order-based nature of your scoring function. Nonetheless, I appreciate that the clarifications will be added to the paper.
>
> All in all, I maintain my rating.

---

> > ### Author Response · Authors · 2024-11-15
> > **Further clarification on symmetry and hierarchy rules**
> >
> > Thank you for taking the time to respond.
> >
> > **Modelling symmetry**
> >
> > We are well-aware of the differences between the inversion and symmetry inference patterns, as studied for instance in the BoxE and ExpressivE papers. Let us try to clarify our point. The original review contains the claim that symmetry cannot be enforced at the relation level. This claim is incorrect if we take into account the fact that inverse triples are added. In the original formulation of the model, a triple $(e,r,f)$ was captured if the following constraint is satisfied:
> >
> > $(\mathbf{B_r}\mathbf{Z_e^{(l)}}\preccurlyeq \mathbf{Z_f^{(l)}}) \wedge (\mathbf{C_r}\mathbf{Z_f^{(l)}}\preccurlyeq \mathbf{Z_e^{(l)}})$
> >
> > where $\mathbf{B_r}$ and $\mathbf{C_r}$ together define the semantics of the relation $r$. With this formulation,  $r$ is symmetric if $\mathbf{B_r}=\mathbf{C_r}$. This situation is entirely analogous to how symmetry is captured in, for instance, ComplEx. In the final formulation of the model, we simplified the definitions and only consider the first constraint, i.e.:
> >
> > $\mathbf{B_r}\mathbf{Z_e^{(l)}}\preccurlyeq \mathbf{Z_f^{(l)}}$
> >
> > The reason is that we can recover the second constraint from the addition of inverse triples, which allows us to keep the formulation of the model simpler, while in terms of expressivity/semantics the situation remains exactly the same.
> >
> > **Modelling hierarchy**
> >
> > We indeed only consider finite rule bases, but it is unclear to us why that would be a practical limitation: we only need infinite rule bases to specify hierarchies if there are infinitely many relations. If we restrict ourselves to finite sets of relations, then arbitrary sets of hierarchy rules can be captured. Indeed, let $r_1,...,r_n$ be the different relations. To construct the rule graph, we create a node $n_0$ and a node $n_{r_i}$ for each relation $r_i$. Then, we add an $r_i$-edge between $n_0$ and $n_{r_j}$ iff the rule base entails $r_i(X,Y)\rightarrow r_j(X,Y)$. It is straightforward to verify that the resulting model will capture the hierarchy rules in our knowledge base (and only these).

---

### Official Review · Reviewer_ugRy · 2024-11-04

**Soundness:** 3
**Presentation:** 3
**Contribution:** 3
**Rating:** 6
**Confidence:** 2

**Summary:**

Reasoning over Knowledge Graphs is an area of machine learning that is primarily concerned with completing knowledge by predicting missing information (relations between entities). This work explores our theoretical understanding of KG embedding approaches that are region-based and studies one aspect of how well these learned embeddings capture rule-like inference patterns. In particular, the authors focus on faithfully capturing closed path reasoning rules and describe a GNN formulation that can provably capture bounded reasoning of such rules. Their proposed Reshuffle model is also efficient, works in the inductive KG completion setting, and performs well empirically on standard KG completion datasets. The authors also show that when a rule exists, there is a formulation of Reshuffle model can faithfully capture these rules.

**Strengths:**

The strength of the work lies in its systematic and formal treatment of question of expressivity of a certain class of KG completion methods. While studying the ability of such models to capture logical rules is not new, the specific type of closed path rules with cyclic dependencies that the authors consider is.

The authors have generally done a good job providing formal proofs for most of their propositions. The paper is well written, with examples that help the reader clarify the definitions and concepts. The definitions are clearly explained (even as notation needs work, see details below) and the work well motivated. In particular, the discussion about negative results in Section 6 is appreciated.

Being able to prove that a model is expressive and has the capacity to capture certain sets of rules is important aspect of research. This work brings us a step closer to understanding and characterizing rule capturing abilities of region-based KG completion models and their limitations.

**Weaknesses:**

There are two main observations I would like to make about this work.

1) Notation: Anyone interested in doing a thorough technical reading of the work will have a hard time keeping track of the notation, which is defined across three sections (3 to 6) as the discussion of ideas moves forward. Given the large span of the use of this notation, it would be better to have a notation section defining most of the key notation so that the reader knows what section to refer to as they read.
In addition, make sure every notation is defined properly (for example, what is an r-edge exactly? I could not find a definition) and avoid overloading notation (for example, on line 370  x^{(l)} means “l repetition of x”, while on the previous page it refers to the message passing layers representations).


2) Cartesian product is mentioned in the abstract as being a limiting factor for many models to capture rules. This mention made me expect more discussion about this later in the paper but there was none. It would be good to expand on how exactly cartesian product ties into this and why is it a limiting factor. There is a relevant paper entitled “Knowledge Hypergraph Embedding Meets Relational Algebra” about capturing of rules (in this case, relational algebra operations) with embedding methods that is missing  from the related work section; this work also highlights the difficulty of capturing cartesian product.



Other remarks and suggestions:
-	The first paragraph of Section 6 mentions that Propositions 1 and 2 imply that Reshuffle can capture rules entailed by P. This is a key result of the paper and needs to be stated formally as a Lemma that follows from these propositions.

-	On line 399, should the (4) be (R4)?

-	Equality on Line 352: (x_{l-1}, r_1, r_l) should be (x_{l-1}, r_1, x_l)

-	The discussion of the results at the end of Section 7 mentions the efficiency of the Reshuffle model compared to NBFNet and GraIL. This efficiency is an important property of this work and should be given a more proper treatment. A quick glance at Table 2 suggests that Reshuffle does not outperform NBFNet in raw performance metrics, highlighting its efficiency benefits would provide valuable context to interpret these.

**Questions:**

Can you give some more insight as to how cartesian product ties into the ability of these models to capture inference patterns?

Did you perform any experiments in a controlled setting, using a synthetic knowledge graph generated based on the closed path rules that this work is trying to capture?  This may allow to better understand in what cases the self-loop relation matters.

---

> ### Author Response · Authors · 2024-11-13
>
> We thank the reviewer for their detailed reading of the paper and constructive feedback.
>
> **Weaknesses**
>
> W1. Thanks for the suggestions for improving the readability of the paper. We say that there is an r-edge from node $n_1$ to $n_2$, for a given knowledge graph or rule graph, if there is a triple $(n_1,r,n_2)$. We will clarify this in the paper. We will also change the notations on line 370 to avoid confusion.
>
> W2. The comment about the Cartesian product in the abstract refers to the limitations of coordinate-wise models. We refer to “Cartesian products of two-dimensional regions” to keep the abstract self-contained (as “coordinate-wise model” is not a standard term). The limitations of existing coordinate-wise models are discussed on lines 049-055, lines 090-097 and lines 152-161.
>
> Thanks for the reference to “Knowledge Hypergraph Embedding Meets Relational Algebra”, which we agree should be added to the related work section. Note, however, that the notion of Cartesian Product that is referred to in that paper refers to the Cartesian product of symbolic relations, which does not have a counterpart in our setting, since we only focus on binary relations. The main interest of the Cartesian Product for knowledge hypergraphs is to compute joins, which are needed to generalise the notion of closed path rules to n-ary relations. Similar to BoxE, it seems that their model may not be able to capture such rules.
>
> Thanks for the other suggestions and for catching the typos. For line 399, the reference to (4) should have been to “Proposition 4”.
>
> Indeed, the efficiency of ReshufflE is an important advantage. The key point is that the embeddings for all entities can be computed with a single forward pass of the model (whose time complexity is similar to that of NBFNet). After this single forward pass, we can evaluate the score of a triple $(e,r,f)$ in constant time (if we treat the number of dimensions of the embedding space as a constant). This is different from models such as NBFNet, which need one forward pass for each query.
>
> **Questions**
>
> Q1. Please refer to our earlier clarification on the notion of Cartesian product. The reason why existing coordinate-wise models cannot model arbitrary rules is intuitively because they cannot treat a given relation differently depending on its position in the body of a rule. Using cross-coordinate comparisons allows us to access different representations of the same relation, in different contexts.
>
> Q2. We did not do experiments with synthetic datasets. Our hypothesis is that self-loops are less important in practice because the matrices that are learned are soft approximations of the binary matrices that are assumed in the theoretical analysis.

---

> > ### Comment · Reviewer_ugRy · 2024-11-19
> >
> > Thank you for the response. Adding the clarifications about notation, efficiency and a more intuitive discussion about cartesian product to the paper would help improve its readability. I still think that experiments on a synthetic dataset would further strengthen the results. I maintain my rating and will revisit after going through the remainder of the author responses.

---

### Official Review · Reviewer_UDqZ · 2024-11-10

**Soundness:** 3
**Presentation:** 3
**Contribution:** 3
**Rating:** 6
**Confidence:** 5

**Summary:**

In this work the authors propose a region-based knowledge graph embedding method which they term ReshufflE. In their method, each entity $e \in \mathcal E$ is represented by a vector $\eta(e) \in \mathbb R^d$, and each relation $r \in \mathcal R$ is represented by a region in $\eta(n) \subseteq \mathbb R^{2d}$. The regions are an intersection of half-spaces, and the score function for a triple $(e, r, f)$ is the $\ell^2$ distance of the concatenated vector $e \oplus f \in \mathbb R^{2d}$ from the region $\eta(r)$.

The authors provide extensive theoretical analysis of the representational capacity of their model, including propositions which claim that the model is capable of representing certain cyclic rules, and moreover capable of embedding any triples which require a bounded number of rule applications from the original graph. Finally, the authors present empirical results on a transductive knowledge graph completion task, where their model does not achieve state-of-the-art results but does seem to deliver "best in class" results considering the more efficient inference of embedding-based models.

**Strengths:**

Both the proposed model and the methods of analyzing the representational capacity are novel. I am not familiar with other approaches which are amenable to the extent of theoretical guarantees regarding the representational capacity as presented in this paper.

The clarity is high. The authors do an excellent job motivating their design decisions and presenting the rather technical arguments which draw on a wide range of foundational concepts. I was able to follow their arguments clearly.

**Weaknesses:**

Significance is moderate. As the authors acknowledge, this model does not obtain SOTA results, and therefore would be unlikely to be leveraged in practice. That said, the authors' techniques and analysis suggest that the model is sufficiently expressive to capture rule bases in practical settings, and so the reason for the difference in performance may be (as they stated) that the evaluation sets are not necessarily rich enough to benefit from these rules alone, and would require a "fallback model".

The overall quality of the work is also moderate. I was very pleased with the principled theoretical analysis the authors presented, however I believe there is an error with the definition of a rule graph which makes some of the propositions incorrect. I believe this may be a small fix, however.


Overall, I very much like this work, and I believe the authors will be able to address the questions I have listed below or correct me if I am mistaken in my understanding.

**Questions:**

**Rule Graph Definition:** First, a minor point is that the rule graph $\mathcal H$ is defined not only with respect to a rule-base $\mathcal P$ but also with respect to the original set of relations $\mathcal R$. (Given a rule-base $\mathcal P$ without a set of relations $\mathcal R$ we can't create a rule graph, since, in particular, it needs one edge for each relation.)

The main question I have is related to the definition of the rule graph, specifically (R4). My interpretation of this rule is the following:

First, for a given multi-graph $\mathcal H$, let $S_{(n_1, n_2)}$ be the set of edge paths from $n_1$ to $n_2$.
Now, given $r\in \mathcal R$, let $T_r =\cup_{(n_1, r, n_2) \in \mathcal H} S_{(n_1, n_2)}$.
Then my interpretation of (R4) is that: for each $r \in \mathcal R$, we have some path $(r_1; \ldots; r_q) \in T_r$ such that $P\models r_1(X_1, X_2) \wedge \cdots \wedge r_q(X_q, X_{q+1}) \rightarrow r(X_1, X_{q+1})$.

Note that the premise of (R4) (as written in the paper) is trivially satisfied because for every two nodes connected by an $r$-edge there is a path connecting these nodes in $T_r$ - in particular, $r$ itself, which is why it wasn't stated in my interpretation above. This could become slightly more interesting if we remove $r$ from $T_r$, in which case we would only apply this condition to situations where $r$ is such that every pair of nodes connected by an $r$-edge have a non-trivial path connecting them, but this is not my main concern.

The main issue I have is that I do not see how this condition ensures that "only the rules in $\mathcal P$ are captured". For example, what prohibits a graph with edges $\\{(n_1, r_i, n_1)\\}_{r \in \mathcal R}$ from being a rule graph for any rule base $\mathcal P$?

I think perhaps what was intended was that $T_r = \cap_{(n_1, r, n_2) \in \mathcal H} S_{(n_1, n_2)} \setminus \\{r\\}$.

Without clarification on this point many of the proofs cannot be verified.

**Example 4:** There is a minor error here in that I think it must be intended that $\mathcal P$ is exactly equal to the set

$$\\{r_1(X, Y) \wedge r_2(Y,Z) \wedge r_1(Z,U) \rightarrow r_2(X, U)\\},$$

because if it merely contains such a rule then it may also contain some other rules which makes it no longer impossible to create a rule graph (eg. $$r_1(X,Y) \rightarrow r_2(X,U)$$). Apart from this, unless the definition for the rule graph is clarified I don't see what would prohibit the graph $\mathcal H$ with edges $\{(n_1, r_1, n_1), (n_1, r_2, n_1)\}$ from satisfying the definition as a rule graph for $\mathcal P$.

**eq edges:** The authors introduce the eq edges, presumably for good reason, but they seem to add complexity and I am not sure what purpose they serve. For example:
1. Line 255: An eq-reduced type of a path is mentioned, but surely any "eq-reduced type"is also, itself, a path in the graph, since eq simply returns to itself. Maybe this was meant to handle the fact that the GNN will have a fixed number of layers, and therefore we need to consider paths of a fixed length always, but it is not clear to me.
2. Figure 1: The rule graphs depicted do not even allow for paths with eq in arbitrary positions, actually to my understanding (perhaps I am wrong?) the depicted rule graphs would not allow for any path which includes "eq" more than once, and then only at the last position.

**Some claims are too strong:**
1. In the abstract, the authors argue that the lack of representational capacity for other models "stems from the use of representations that correspond to the Cartesian produce of two-dimensional regions", however the author's proposed model also uses regions which are cartesian products of 2-dimensional regions.
2. On line 153 the authors claim that BoxE is not capable of capturing closed path rules, but this doesn't seem to be true - simply represent $r$ by any box which contains the intersection of the boxes for $r_1, \ldots, r_p$.

---

> ### Author Response · Authors · 2024-11-13
>
> We thank the reviewer for the detailed and constructive feedback. Below we clarify some misunderstandings, which we believe will address the concerns that were raised.
>
> **Rule graph definition**
>
> Indeed, we implicitly assume that $\mathcal{R}$ is the set of relations that appear in $\mathcal{P}$. We will clarify this point.
>
> Condition (R4) is stronger than the interpretation given by the reviewer. The correct interpretation, using the notation suggested by the reviewer, is as follows:
>
> Let $\\{(r_{11};...;r_{1p_1}), ..., (r_{q1};...;r_{qp_q})\\}$ be any set of paths such that $S_{n_1,n_2} \cap \\{(r_{11};...;r_{1p_1}), ..., (r_{q1};...;r_{qp_q})\\} \neq \emptyset$ holds for every pair of nodes $(n_1,n_2)$ connected by an r-edge.  Then we must have $\mathcal{P}\models r_{i1}(X_1,X_2)\wedge ... \wedge r_{ip_i}(X_{p_i},X_{p_{i+1}})\rightarrow r(X_1,X_{p_{i+1}})$ for some $i\in \\{1,...,q\\}$.
>
> It is true that the premise is trivially satisfied for the set $\{r\}$, and for that specific choice the conclusion is also trivial, since we always have $\mathcal{P}\models r(X_1,X_2)\rightarrow r(X_1,X_2)$. However, (R4) has to be satisfied for **all** sets of paths that satisfy the condition. We will clarify this point in the paper.
>
> To see why (R4) is needed, assume that for any pair of nodes $(n_1,n_2)$ connected by an r-edge, there is a path $r_1;r_2$. Then the corresponding model would infer $r(a,c)$ whenever the knowledge graph contained the triples $(a,r_1,b)$ and $(b,r_2,c)$ for some entities $a,b,c$. So such rule graphs should only be allowed if $r_1(X_1,X_2) \wedge r_2(X_2,X_3) \rightarrow r(X_1,X_3)$ is entailed by $\mathcal{P}$. However this is not enough. Suppose that there are two r-edges in the rule graph, say between $n_1$ and $n_2$ and between $n_3$ and $n_4$. Suppose also that there is an $r_1;r_2$ path connecting $n_1$ and $n_2$, and an $r_3;r_4$ path connecting $n_3$ and $n_4$. Now suppose the knowledge graph contained the triples $(a,r_1,b),(b,r_2,c),(a,r_3,d),(d,r_4,c)$, then the corresponding model would infer $(a,r,c)$. In other words, the model would capture the rule $r_1(X_1,X_2)\wedge r_2(X_2,X_3)\wedge r_3(X_1,X_4)\wedge r_4(X_4,X_3)\rightarrow r(X_1,X_3)$, but such a rule can never be entailed from $\mathcal{P}$, since the latter only contains closed path rules. So we need to make sure that such rule graphs are only allowed if $r_1(X_1,X_2)\wedge r_2(X_2,X_3)\rightarrow r(X_1,X_3)$ or $r_3(X_1,X_4)\wedge r_4(X_4,X_3)\rightarrow r(X_1,X_3)$ is entailed from $\mathcal{P}$, which is precisely what (R4) ensures. We will clarify this point in the paper.
>
> **Example 4**
>
> Indeed, the intention is that $\mathcal{P}$ is equal to the given singleton. We will clarify this.
> Choosing $\mathcal{H}$ with edges $(n_1,r_1,n_1)$ and $(n_1,r_2,n_1)$ wouldn’t satisfy condition (R4). For instance the set of rule paths $\\{r_1;r_2\\}$ satisfies the premise of (R4) for $r=r_2$. Indeed, any pair of nodes connected by an $r_2$ edge is also connected by an $r_1;r_2$ path, yet $\mathcal{P}$ does not entail the rule $r_1(X_1,X_2)\wedge r_2(X_2,X_3)\rightarrow r_2(X_1,X_3)$.
>
> **eq edges**
>
> The eq-relation connects entities with themselves in the knowledge graph, but in the rule graph, there can be eq-edges between different nodes, so the eq-reduced type of a path may itself not be a path.
>
> Indeed, the eq-edges play no meaningful role in the examples illustrated in Figure 1. However, in general, when constructing rule graphs, eq-edges are sometimes needed to make sure that condition (R2) can be satisfied. Figure 2 (in the appendix) gives an example of such a case. In that example, we would like to have an $r_2$-edge from $n_{r_4}$ to $n_1$ and an $r_2$-edge from $n_{r_5}$ to $n_1$, but then there would be two incoming $r_2$-edges in $n_1$, meaning that (R2) is violated. This is addressed by creating the node $n_2$, adding an $r_2$-edge from $n_{r_5}$ to $n_2$ and an eq-edge from $n_2$ to $n_1$.
>
> **Some claims are too strong**
>
> The regions that are used in our model cannot, in general, be expressed as the Cartesian product of two-dimensional regions. The sentence from the abstract refers to the fact that existing region-based models compare entity embeddings component-wise. In the proposed approach, we lift this limitation, allowing any coordinate from the representation of the head entity to be compared with any coordinate from the representation of the tail entity.
>
> In the case of BoxE, choosing $r$ such that it contains the intersection of $r_1,...,r_p$ would capture the following rule:
>
> $r_1(X,Y)\wedge … \wedge r_p(X,Y) \rightarrow r(X,Y)$
>
> Which is an “intersection rule” rather than a closed path rule. BoxE can indeed capture arbitrary sets of intersection rules, but it is provably incapable of capturing closed path rules. Note that this limitation is acknowledged in the BoxE paper (e.g. in their Table 1, where closed path rules are referred to as “composition rules”).

---

### Meta-Review · Area_Chair_zwus · 2024-12-23

**Metareview:**

Summary: The paper proposes a region-based knowledge graph embedding model designed to address limitations in capturing various inference rules. The model uses ordering constraints to encode relations, extending the capability of existing models to handle a broader class of closed path rules, including some with cyclic dependencies. Embeddings are learned using a Graph Neural Network (GNN), enabling efficient updates and improved reasoning. The model theoretically guarantees bounded inference for arbitrary rule sets and demonstrates empirical efficiency on inductive knowledge graph completion tasks.

Strength:
- Proposed model and methods are novel
- Systematic and formal treatment of expressivity/representational capacity

Weakness:
- Lukewarm responses from the reviewers
- Over-emphasis on expressing closed path rules, at the expense of, e.g., symmetry at the expense of other desirable properties and no analysis for understanding the trade-offs are presented
- Some of the claims are too strong
- Limited empirical evaluations, e.g. missing baselines, not finding complex real datasets where the proposed method can shine and
- Practical implications questioned by reviewers: As the model does not obtain SOTA results or can handle noisy data, etc.
- Presentation and organization Issues

Decision:
While the work shows promise and provides an interesting analysis and is a borderline case. We acknowledge the primary theoretical nature of the work, but as reviewers pointed out to make readers appreciate the importance of expressing closed path rules theory and the theoretical analysis, better trade-off needs to be presented (theory or empirical).  Addressing all the concerns would warrant another round of reviewing.

**Additional Comments On Reviewer Discussion:**

We thank the authors and reviewers for engaging during the discussion phase towards improving the paper. Almost all reviewers question the over-emphasis on one property. Below are some of the other highlights:

1. Rule Graph Definition:
- Reviewer found potential issues with rule graph definition
- Authors clarified the definitions and explained how conditions prevent undesired rule captures
- Response was technically sound and addressed the concerns

2. Model Capabilities:
- Questioned handling of symmetry and hierarchy rules
- Authors explained how these can be captured through inverse relations and graph construction, but reviewers not totally convinced.

3. Experimental Validation:
- Requested more comprehensive experiments and baselines
- Authors explained focus on theoretical contributions while acknowledging empirical limitations

4. Notation and Presentation:
- Suggested improvements to notation and explanations
- Authors agreed to clarify definitions and improve presentation

In summary, though the authors provided good clarifications in rebuttal, many suggested improvements would require substantial revision beyond what's possible in the rebuttal period.

---

### Decision · Program_Chairs · 2025-01-22

Reject